# Biosensor and machine learning-aided engineering of an amaryllidaceae enzyme

Simon d'Oelsnitz [1,6] ✉, Daniel J. Diaz [2,3], Wantae Kim[4], Daniel J. Acosta [1], Tyler L. Dangerfield[1], Mason W. Schechter[1], Matthew B. Minus[5], James R. Howard[2], Hannah Do[1], James M. Loy [1], Hal S. Alper [4], Y. Jessie Zhang [1] & Andrew D. Ellington [1]

A major challenge to achieving industry-scale biomanufacturing of therapeutic alkaloids is the slow process of biocatalyst engineering. Amaryllidaceae alkaloids, such as the Alzheimer's medication galantamine, are complex plant secondary metabolites with recognized therapeutic value. Due to their difficult synthesis they are regularly sourced by extraction and purification from the low-yielding daffodil *Narcissus pseudonarcissus*. Here, we propose an efficient biosensor-machine learning technology stack for biocatalyst development, which we apply to engineer an Amaryllidaceae enzyme in *Escherichia coli*. Directed evolution is used to develop a highly sensitive ($EC_{50} = 20\ \mu M$) and specific biosensor for the key Amaryllidaceae alkaloid branchpoint 4'-O-methylnorbelladine. A structure-based residual neural network (MutComputeX) is subsequently developed and used to generate activity-enriched variants of a plant methyltransferase, which are rapidly screened with the biosensor. Functional enzyme variants are identified that yield a 60% improvement in product titer, 2-fold higher catalytic activity, and 3-fold lower off-product regioisomer formation. A solved crystal structure elucidates the mechanism behind key beneficial mutations.

Alkaloids produced by the *Amaryllidoideae* subfamily of flowering plants have great therapeutic promise, including anticancer, fungicidal, antiviral, and acetylcholinesterase inhibition properties. Among the approximate ~600 reported *Amaryllidoideae* alkaloids (AAs), those derived from the lycorine, haemanthamine, and narciclasine scaffolds have been used as lead molecules in anticancer research[1–4]. One of the most notable AAs is galantamine, a selective and reversible acetylcholinesterase inhibitor that is a licensed treatment for mild to moderate symptoms of Alzheimer's disease and a promising scaffold for drug design[5,6]. Due to galantamine's challenging synthesis, global supplies largely rely on isolating the low quantities (0.3% dry weight)

that accumulate in harvested daffodils, ultimately resulting in an expensive ($50,000/kg) and environmentally-dependent supply chain[7,8]. In an effort to improve galantamine production, agricultural techniques are currently being tested to boost daffodil-sourced yields[9,10].

A promising alternative to amaryllidaceae alkaloid extraction from plants is microbial fermentation. Recently, long plant pathways have been reconstituted into microbial hosts for the production of therapeutic benzylisoquinoline alkaloids[11,12], tropane alkaloids[13], and monoterpene indole alkaloids[14]. While the complete biosynthetic pathway for any AA with therapeutic value has not yet been elucidated,

[1]Department of Molecular Biosciences, University of Texas at Austin, Austin, TX 78712, USA. [2]Department of Chemistry, University of Texas at Austin, Austin, TX 78712, USA. [3]Institute for Foundations of Machine Learning, University of Texas at Austin, Austin, TX 78712, USA. [4]McKetta Department of Chemical Engineering, University of Texas at Austin, Austin, TX 78712, USA. [5]Department of Chemistry, Prairie View A&M University, 100 University Dr, Prairie View, TX 77446, USA. [6]Present address: Synthetic Biology HIVE, Department of Systems Biology, Harvard Medical School, Boston, MA 02115, USA. ✉e-mail: simonsnitz@gmail.com

recent studies have characterized early pathway enzymes responsible for the biosynthesis of 4'-O-Methylnorbelladine, the last common intermediate before AA pathway branches diverge[15]. Furthermore, semi-synthetic methods have been proposed using characterized enzymes to generate advanced intermediates[16]. The industrial application of such pathways could be greatly accelerated by augmenting high-throughput screens with genetic biosensors[17–20], and using machine learning to guide protein design[21–24], yielding enzymes and pathways with improved stability and activity.

Here, we synergize the development of custom biosensors with machine learning(ML)-guided protein design to improve microbial fermentation of the branchpoint AA 4'-O-methylnorbelladine (4NB). A generalist transcription factor, RamR, is evolved into a highly sensitive biosensor for 4NB that precisely discriminates against the non-methylated precursor norbelladine, and the biosensor is then used to monitor the activity of norbelladine 4'-O-methyltransferase (Nb4OMT) from the daffodil *Narcissus pseudonarcissus* in *Escherichia coli*. We then develop MutComputeX: a structure-based self-supervised residual neural network (3DResNet) trained to generalize at protein:non-protein interfaces, which is used to generate activity-enriched Nb4OMT designs from an ML-generated protein-cofactor-substrate structure. The evolved biosensor is used to rapidly screen a panel of MutComputeX-guided Nb4OMT designs, leading to the identification of one variant that yields a 60% improvement in product titer, 2-fold higher catalytic activity, and 3-fold lower off-product formation. A newly solved crystal structure of this engineered enzyme helps elucidate the mechanism behind key beneficial mutations and highlight important discrepancies with the AlphaFold2 model.

## Results

### Identifying a biosensor for the branchpoint amaryllidaceae alkaloid 4'-O-methylnorbelladine

4'-O-methylnorbelladine (4NB) is the branchpoint intermediate for the entire amaryllidaceae alkaloid (AA) family (Fig. 1a), and therefore was the target compound for biosensor generation. Previously the highly malleable TetR-family *Salmonella typhimurium* repressor RamR had been used as a starting point for identifying biosensors for a variety of benzylisoquinoline alkaloids[17]. Given the chemical similarities between AAs and BIAs, and RamR's proven ability to rapidly evolve novel ligand specificity, RamR was again used as a starting point for directed evolution.

The wild-type RamR sensor was constitutively expressed on one plasmid (pReg-RamR) in parallel with another plasmid bearing the regulator's cognate promoter upstream of the sfGFP gene (Pramr-GFP). Upon induction with various AA intermediates, RamR was found to be slightly responsive to both 4NB and its immediate precursor norbelladine, yielding 3.8-fold and 4.4-fold increases in fluorescence, respectively (Fig. 1b). To better understand this promiscuous binding activity, 4NB was docked within the ligand-binding pocket of RamR using GNINA 1.0[25], and a conformational pose was identified whereby the phenol moiety of norbelladine forms hydrogen bonds with S137 and T85, while the catechol moiety forms a hydrogen bond with K63. This docking pose also suggested that norbelladine's secondary amine may hydrogen bond with D152 and further interact with the aromatic ring of F155 (Fig. 1c).

### Evolving a highly specific biosensor for 4'-O-Methylnorbelladine

While the native responsiveness was promising, for practical use in metabolic engineering applications the sensitivity and specificity of RamR for 4NB needed to be greatly improved. The simulated molecular interactions between RamR and 4NB informed a rational approach to library design. Three site-saturated (NNS) RamR libraries that each targeted three residues facing inwards toward the ligand-binding cavity were generated (Supplementary Fig. 1; see "Methods"). The 32,000 unique genotypes per library could be readily plumbed using our previously described method, Seamless Enrichment of Ligand Inducible Sensors (SELIS)[17]. Briefly, this method involves a growth-based selection to first filter out biosensor variants that are incapable of repressing transcription from their cognate promoter, followed by a fluorescence-based screen to isolate sensor variants highly responsive to the target analyte.

After the first round of directed evolution, several RamR variants were found to be substantially more responsive to 4NB, even in the absence of a negative selection against norbelladine. In fact, one variant bearing two amino acid substitutions (4NB1.2: K63T and L66M) displayed a 20-fold selectivity for 4NB over norbelladine (Supplementary Fig. 2a, b). Although two other RamR variants had greater sensitivity for 4NB, the higher selectivity of the 4NB2.1 variant rendered it more suitable for accurately monitoring pathway activity. Using 4NB1.2 as a starting point, additional libraries were generated that encompassed the other, previously randomized positions. SELIS was now performed with a growth-based counter-selection against

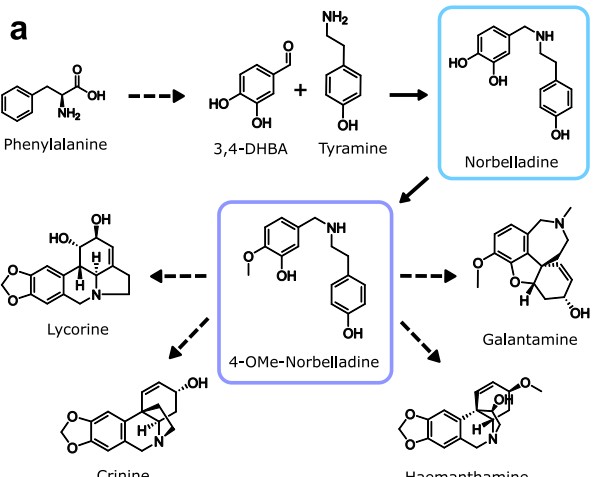

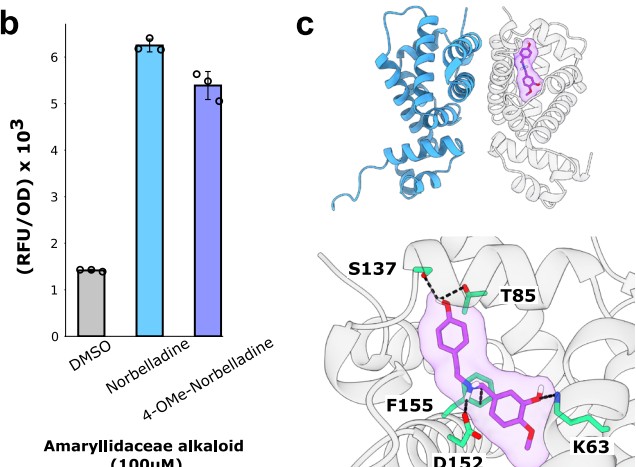

**Fig. 1 | Identifying a biosensor responsive to amaryllidaceae alkaloid intermediates. a.** Abbreviated biosynthetic plant pathways for therapeutic amaryllidaceae alkaloids. **b** Response of the RamR transcription factor to amaryllidaceae alkaloid pathway intermediates norbelladine and 4'-O-methylnorbelladine. Error bars represent the S.D. +/− the mean. Experiments were conducted in biological triplicate. **c** Structure of RamR (PDB: 3VVX_A) docked with 4'-O-methylnorbelladine using GNINA (see "Methods"). Predicted ligand-interacting residues are highlighted green.

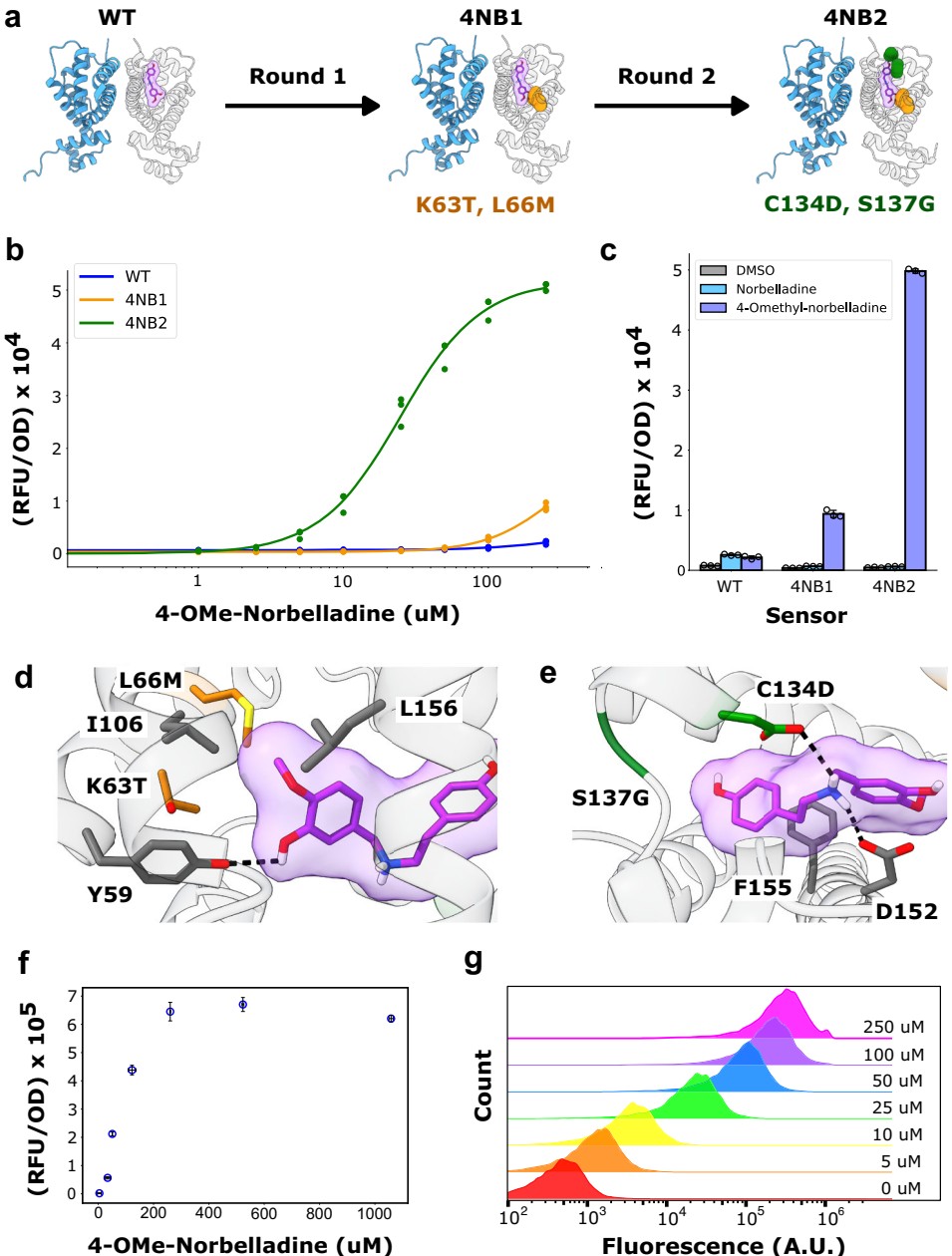

**Fig. 2 | Evolving a highly specific biosensor for 4′-O-methylnorbelladine.**
**a** Schematic illustrating the mutations that resulted after round one (4NB1) and round two (4NB2) of RamR evolution towards 4′-O-methylnorbelladine. **b** Dose–response measurements of WT RamR, 4NB1, and 4NB2 mutants with 4′-O-methylnorbelladine. **c** Relative response of WT RamR, 4NB1, and 4NB2 mutants to norbelladine and 4′-O-methylnorbelladine. **d**, **e** AlphaFold2 structural model of 4NB2 docked with 4′-O-methylnorbelladine. Predicted ligand interactions with WT residues, mutations that arose in 4NB1, and mutations that arose in 4NB2 are colored gray, orange, and green, respectively. (**f**) Correlation between fluorescent response measured with the 4NB2 sensor and 4′-O-methylnorbelladine measured with high-performance liquid chromatography. (**g**) The distribution of fluorescent cell populations in response to 4′-O-methylnorbelladine concentration. All data was performed in biological triplicate. Cells were cultured for 4 h with the ligand in (**b**) and (**c**), and for 18 h with the ligand in (**f**) and (**g**). Error bars represent the S.D. +/– the mean.

norbelladine (100 μM). The top four biosensor variants were again highly specific for 4NB but now also became significantly more sensitive, with the best variant, 4NB2.1 (C134D and S137G), achieving a limit of detection of approximately 2.5 μM (Supplementary Fig. 2c, d; Fig. 2). Ultimately, the 4NB2.1 sensor was highly selective for 4NB over norbelladine, displaying an over 80-fold preference for the former, despite the two effectors differing by only a single methyl group.

To again explore the structural basis for precise methyl group discrimination a structural model of 4NB2.1 was generated using AlphaFold2[26], and 4NB was docked into this model using GNINA 1.0[25].

The docked pose suggests that the K63T substitution repositions the hydroxyl group at position 3 of 4NB to hydrogen bond with the wild-type Y59 residue, while the L66M substitution strengthens a hydrophobic pocket around the 4′-O-Methyl group of 4NB (along with the native I106 and L156 residues; Fig. 2d). This analysis is in agreement with the fluorescence assay data, since only RamR variants bearing the K63T and L66M mutations are highly selective for 4NB over norbelladine (Supplementary Fig. 2). The model also positions the new aspartate at position 134 (C134D) to hydrogen bond with the amine of 4NB; several other RamR variants also placed a hydrogen bond donor

(glutamate, glutamine, asparagine) at the 134 position (Supplementary Fig. 2). Overall, as a consequence of these substitutions the 4NB ligand may shift in position to allow for more favorable π–π stacking with F155 (Fig. 2e).

To evaluate the utility of the 4NB2.1 sensor for high-throughput screening of AA intermediates, we compared its performance to an HPLC method adapted from the literature[27]. The concentration range of 4NB can be discerned between 2.5 μM and 250 μM, while the equivalent range for the HPLC method is between 25 μM and 1000 μM (Fig. 2f, Supplementary Table 1). The dynamic range of sensing could potentially be further increased via less sensitive biosensor intermediates characterized during evolution (see Supplementary Fig. 2). Most importantly, the 4NB2.1 sensor is approximately 10-fold more sensitive than the HPLC method, making it well-suited for screening transplanted biosynthetic enzymes from plants, which often initially show low flux[28]. Flow cytometry analysis indicated that the sensor's response at the population level was highly uniform (Fig. 2g), ensuring low noise measurements.

## Monitoring norbelladine methyltransferase activity in *Escherichia coli*

Although several AAs have been recognized for their therapeutic value, to our knowledge there have so far been no attempts to reconstitute AA pathways in microbial hosts. Since norbelladine 4′-O-methyltransferase (Nb4OMT) from the wild daffodil *Narcissus sp. aff. pseudonarcissus*, is directly responsible for 4NB production from norbelladine, we chose this as a starting point for development of a fuller pathway. A 4NB reporter plasmid (pSens4NB2; Supplementary Fig. 3) was co-transformed with a plasmid constitutively expressing Nb4OMT. When this strain was grown in media supplemented with the substrate norbelladine, Nb4OMT activity could be observed, monitored, and quantified via fluorescence (Fig. 3a). The level of cell fluorescence correlated positively with enzyme expression strength (Supplementary Fig. 4), with the concentration of norbelladine supplemented into the culture media, and with 4NB titer measured via HPLC (Fig. 3b). As was the case with measuring 4NB supplemented media, the fluorescence of cellular populations was uniformly distributed, again indicating that there was little noise during production or sensing. The independent measurements of noise via the 4NB biosensor will likely prove important as high yield strains are further developed and translated.

While these results demonstrated the utility of the evolved biosensor for monitoring Nb4OMT activity, they also revealed the catalytic inefficiency of the enzyme. HPLC analysis indicated that a significant amount of supplemented norbelladine remained after culturing for 24 h (Fig. 3c). Indeed, leftover norbelladine was identified when as low as 50 μM of norbelladine was supplemented into the culture media (Supplementary Fig. 5). Furthermore, LC/MS analysis identified 3′-O-Methylnorbelladine as a minor component, indicating that the wild-type Nb4OMT enzyme was not highly regiospecific (Supplementary Fig. 6). These observations all suggested that Nb4OMT activity and specificity could be improved by enzyme engineering.

To improve Nb4OMT activity in a microbial host we initially carried out directed evolution starting from randomly mutagenized libraries, via error-prone PCR, which generated an average of three mutations per gene. The library of enzyme variants was transformed into cells containing the pSens4NB2.1 plasmid, plated on solid media containing norbelladine, and highly fluorescent colonies were isolated and then individually phenotyped in a secondary, quantitative liquid-based fluorescence screen where they were compared to the wild-type enzyme. Unfortunately, this approach was not able to identify variants that outperformed the wild-type enzyme in the liquid-based screen.

## Developing a machine learning pipeline for structure-based enzyme engineering

To pursue a complementary approach to enzyme engineering, we sought to use machine learning to guide enzyme design, an approach that could identify variants unlikely to occur via random mutagenesis. MutCompute is a self-supervised convolutional neural network (CNN) trained to use a local 3D chemical microenvironment to predict amino acid likelihood at each residue within a protein. We have previously demonstrated that positions where MutCompute does not predict the wild-type amino acid can frequently be substituted with a more chemically congruent amino acid, which has enabled us to improve protein fluorescence (BFP)[29], expression (PMI)[29], stability (polymerase)[30], and catalytic activity (PETase)[21].

The original data engineering pipelines established for MutCompute restricted its training to microenvironments with atoms belonging to the 20 amino acids, and therefore MutCompute was unable to provide contextualized predictions in microenvironments that possessed atoms from cofactors, ligands, or nucleic acids[29,31]. To create designs that could be generalized to protein-ligand interfaces, we developed MutComputeX: an improved structured-based neural network designed for protein engineering. (Fig. 4a). To develop MutComputeX, we first rebuilt the data engineering pipelines to enable training on heterogenous microenvironments (Fig. 4b). New atomic channels for phosphorus and grouped halogens were added to the input representation (see "Methods"). New training and testing datasets were curated that included sampling ~256,000 protein-ligand interface microenvironments (see "Methods"). Finally, a residual convolutional architecture was developed to improve feature extraction capabilities and in turn the predictive power of the model[32,33] (Fig. 4c, Supplementary Fig. 7). The self-supervised 3D residual neural network (3DResNet) framework achieved an improved wild-type prediction accuracy of ~80% on a ~250 K residue test set compared to 69% on a 6 K test set from the previous 3DCNN model[29,31]. Furthermore, the 3DResNets were shown to generalize to protein-ligand interaction interfaces without any drop in wild-type accuracy (81% wild-type accuracy on a protein-ligand interface test set compared to 62.1% from the previous 3DCNN model). After training numerous models, we selected three models for ensembling and ML-engineering of the norbelladine methlytransferase based on their zero-shot capability to correlate with ΔTM point mutations from FireProtDB[34] (zero-shot correlation described in "Methods"). The ensembled 3DResNet model (MutComputeX) had an overall wild-type accuracy of 67.3% and protein-ligand interface wild-type accuracy of 66.%. While the wild-type accuracy of MutComputeX is lower than what is capable by the 3DResNet framework, we chose to use this model since it correlated best with experimentally collected data from FireProtDB[34].

To produce MutComputeX-guided designs, we generated a Nb4OMT enzyme structure file to serve as an input to the model. Although the structure of the Nb4OMT enzyme had not been solved, we were able to create a de novo structural model for Nb4OMT using AlphaFold2[26], which was then docked with both the S-adenosylhomocysteine (SAH) cofactor and norbelladine using GNINA1.0[25]. The SAH cofactor was chosen instead of SAM because the nearest structure, of Alfalfa caffeoyl coenzyme A 3′-O-methyltransferase (PDB: 1SUI; sequence similarity: 60.79%), contained this cofactor, and its SAH pose was transplanted to the AlphaFolded Nb4OMT scaffold. GNINA scored the minimized SAH pose with a 0.835 probability of being within 2 Å RMSD from the real pose, and predicted an affinity of −7.9 kcal/mol (Supplementary Table 2). The GNINA pose was guided by the supposition that either D155 or K158 must be the general-base that deprotonates the 4-hydroxyl group during the SN2 reaction, and that a potential cation-π interaction with K158 would orient the plane of the catechol ring in the active site. GNINA scored the minimized norbelladine pose with a 0.824 probability of being within 2 Å RMSD from the

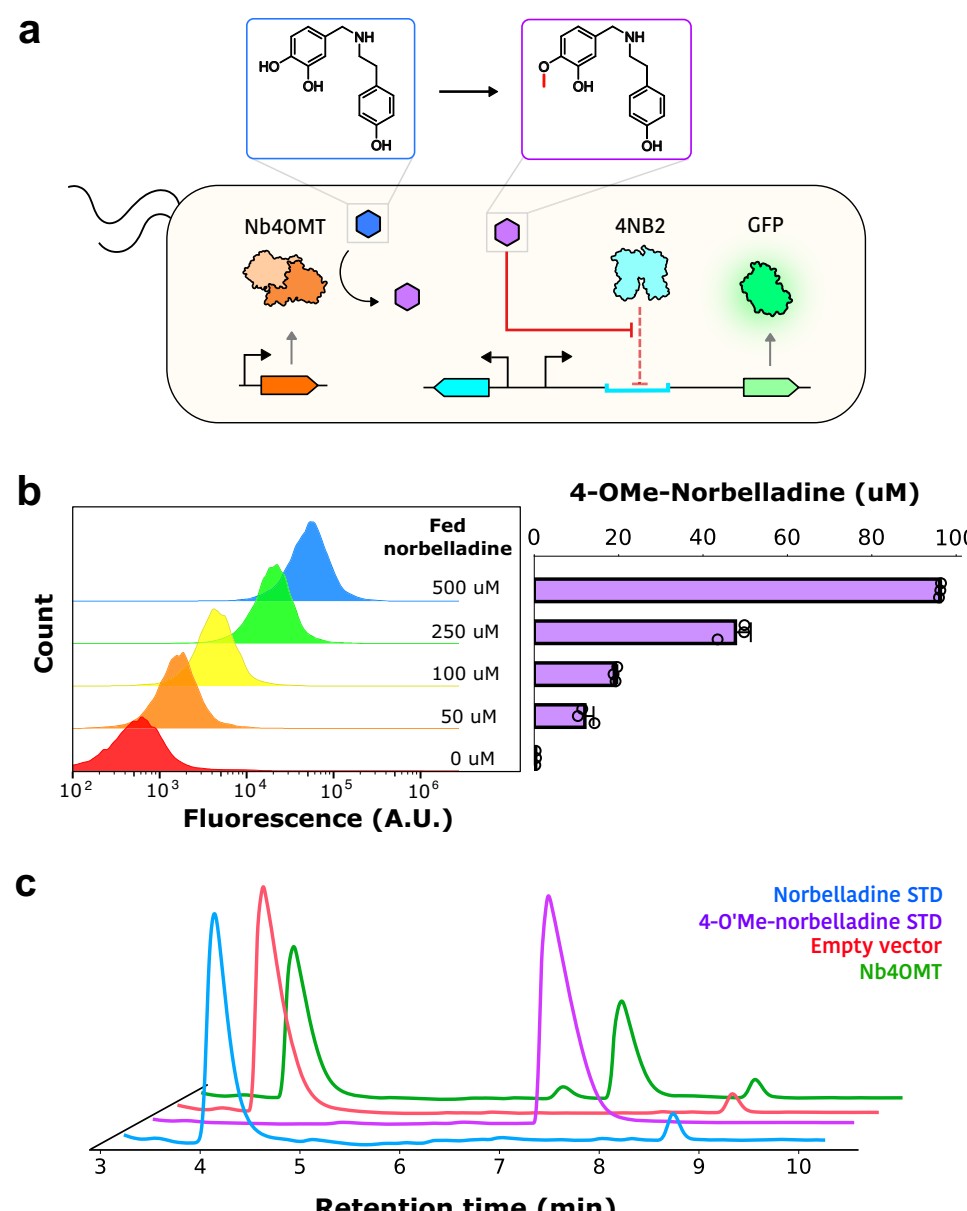

**Fig. 3 | Monitoring Nb4OMT activity with the 4NB2 biosensor. a** Schematic representation of the biosensor-monitored enzymatic reaction within *E. coli* cells. Blue and purple hexagons denote norbelladine and 4′-O-methylnorbelladine, respectively. The schematic was designed using a vector graphics editor. **b** Correlation between cell population fluorescence and biosynthesized 4′-O-methylnorbelladine, measured by high-performance liquid chromatography. All data was performed in biological triplicate. Error bars represent S.D. +/− the mean. **c** Chromatographic traces of supernatant collected from *E. coli* cells expressing an active Nb4OMT enzyme (green) or an empty plasmid (red). Traces for norbelladine and 4′-O-methylnorbelladine standards are shown in blue and purple, respectively.

real pose and a predicted affinity of −7.3 kcal/mol (Supplementary Table 2). The ternary Nb4OMT structure model was passed to Mut-ComputeX, and predictions were generated for each residue on both chains (Fig. 4d). Based on these predictions, we manually curated predicted substitutions, prioritizing those that were near the active site and that were likely to form known stabilizing motifs, such as salt bridges. Our rationale for choosing the selected mutations is provided in Supplementary Discussion 1.

### Characterization of improved norbelladine methyltransferase variants

Ultimately, 22 mutational designs were experimentally validated in *E. coli*. Leveraging the biosensor-enabled high-throughput screen, we were able to quickly assess each of the 22 mutants across three temperatures (25 °C, 30 °C, 37 °C) and two substrate concentrations

(100 μM, 1 mM) (Supplementary Fig. 8). In all tested conditions, the A53M mutation consistently produced a fluorescent signal significantly above the wild-type enzyme, while the H17K, H17R, S159E, V203E, and E36P-G40E substitutions produced signals above wild-type in at least one tested condition (Supplementary Fig. 8). Increasing the reaction temperature to 37 °C improved product formation (despite the fact that the *Narcissus pseudonarcissus* plant grows in 10–23 °C climates[35]). Double and triple mutants incorporating the H17K, A53M, S159E, V203E, and E36P-G40E substitutions were generated and screened; as with the initial screens, variants bearing the A53M mutation produced the greatest signals (Fig. 5a). In time course reactions, after media supplementation with norbelladine the rate of fluorescence increase for the E36P-G40E variant was similar to that of the wild-type enzyme, but the rates produced by the two A53M-bearing variants were significantly higher (Fig. 5b). LC/MS analyses were

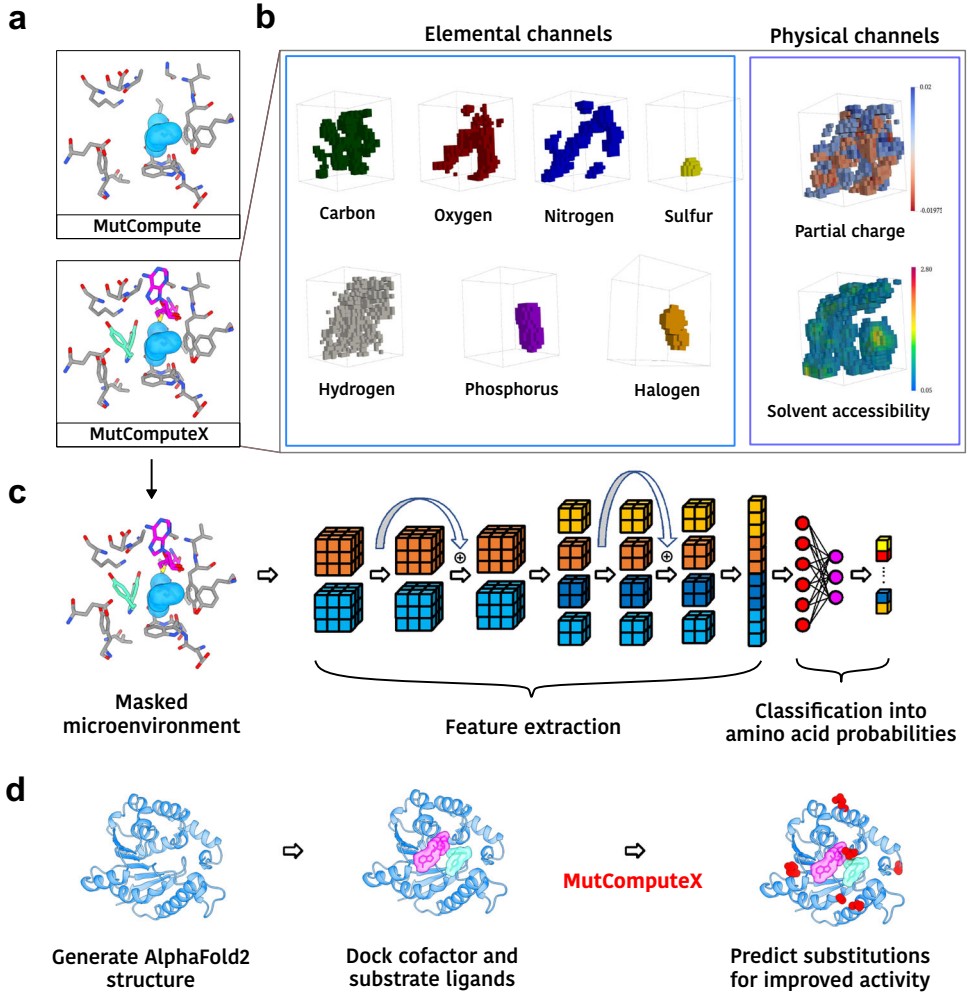

**Fig. 4 | The MutComputeX pipeline. a** The A53 microenvironment inputs for MutCompute (top), which does not include non-protein atoms, and for MutComputeX (bottom), which includes both ligand and cofactor atoms (PDB: 8UKE). **b** The microenvironment is voxelized into seven elemental and two physical channels. All halogen atoms are combined into a single channel. **c** An overview of MutComputeX residual neural network architectures. A more detailed architecture diagram is provided in Supplementary Fig. 7. The schematic was designed using a vector graphics editor. **d** Workflow using MutComputeX for enzyme engineering. In the A53 masked microenvironment that is shown, the light blue spheres represent the masked alanine, the norbelladine ligand is shown in aqua, protein residues are shown in gray, and S-adenosyl-homocysteine (SAH) is shown in pink.

carried out on supernatants from the E36P-G40E, A53M, and E36P-G40E-A53M variants, and in agreement with our fluorescence-based assay the level of 4NB product increased by 60% while the level of remnant norbelladine decreased 17-fold (Fig. 5c, Supplementary Fig. 9). Furthermore, the A53M mutation reduced levels of the 3′-O-Methyl-norbelladine off-product by about 3-fold (Supplementary Fig. 10). Interestingly, we found that the beneficial A53M substitution was only predicted by MutComputeX when the Nb4OMT structure model was docked with SAH and norbelladine; in contrast, A53R was predicted when docking was not performed, a substitution that reduced activity under all tested conditions (Supplementary Fig. 8, Supplementary Fig. 11). These results clearly demonstrate that the incorporation of ligand atoms to the microenvironment greatly improves MutComputeX's ability to engineer the active site of enzymes.

To further understand the mechanism behind beneficial mutations, we characterized the steady state kinetic and thermal properties of NbOMT bearing the A53M substitution alone or in combination with the E36P and G40E substitutions. The A53M substitution increased $k_{cat}/K_m$ by a factor of about 2, due to a > 2.1-fold increase in $k_{cat}$, and increased the $T_m$ by 1.7 °C relative to the wild-type enzyme (Table 1; Supplementary Fig. 12). The Nb4OMT$^{E36P/G40E/A53M}$ triple substitutions

appeared to have $k_{cat}$ and $K_m$ values similar to the Nb4OMT$^{A53M}$ single mutant, but a 5.6 °C increase in $T_m$ relative to the wild-type Nb4OMT enzyme. Steady state kinetic data suggested that the Nb4OMT$^{A53M}$ and Nb4OMT$^{E36P/G40E/A53M}$ mutant enzymes were affected by substrate inhibition (Supplementary Fig. 12). These in vitro characterization data agree with the in vivo data collected with the 4NB-responsive biosensor (Fig. 5a).

## Crystal structure of an improved norbelladine methyltransferase

To better understand the mechanism underlying the three beneficial substitutions in the Nb4OMT$^{E36P/G40E/A53M}$ variant, we determined the structure of the Nb4OMT$^{E36P/G40E/A53M}$ variant in complex with S-adenosyl-L-homocysteine (SAH) at 2.4 Å resolution. The Nb4OMT variant exists as a homodimer in the crystalline form (Fig. 6a), consistent with its size exclusion chromatogram (Supplementary Fig. 13). The overall fold of the protein was almost identical to the predicted AlphaFold2 structure, except for the N-terminal region (Fig. 6b). AlphaFold2 predicts that Lys13 forms tight salt bridge interaction with Asp155, Asp181, and Asn182 in the enzyme active site, while the experimental structure showed that Asp155, Asp181, and Asn182 instead coordinate a Ca$^{2+}$ ion and Lys13

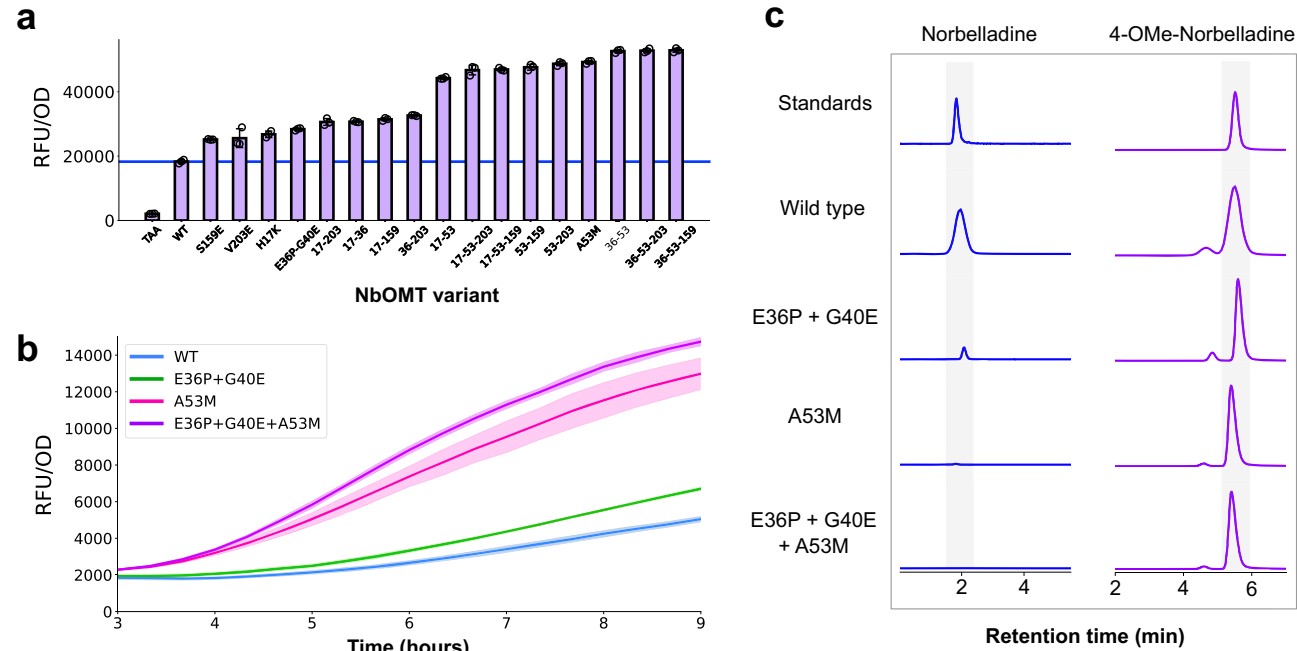

**Fig. 5 | In vivo characterization of ML-designed Nb4OMT variants. a** Fluorescent signal produced from *E. coli* cells containing the 4'-O-methylnorbelladine reporter plasmid (pSens-4NB2) and expressing either an empty plasmid (TAA), the wild-type Nb4OMT enzyme (WT), or Nb4OMT mutants, when cultured with 100 µM of norbelladine at 37 °C. The blue horizontal line denotes the fluorescent signal produced from culturing the wild-type Nb4OMT enzyme. All data was performed in biological triplicate. Error bars represent the S.D. +/− the mean. Genotypes of all variants can be found in Supplementary Table 3. **b** Time-dependent fluorescent signals produced by *E. coli* cells containing the 4'-O-methylnorbelladine reporter plasmid (pSens-4NB2) and expressing Nb4OMT (WT) or ML-designed mutants. Data was performed in biological triplicate and shaded error bands represent the S.D. +/− the mean. **c** Ion-extracted chromatograms of chemical standards (blue) or the supernatant of cells expressing Nb4OMT or ML-designed mutants (purple) cultured with norbelladine.

forms hydrogen bonds with the backbone of Tyr186 and the side-chain of Tyr194 (Fig. 6b).

The experimental structure of Nb4OMT[E36P/G40E/A53M] provides a basis for the improved thermostability of the enzyme (an increase in $T_m$ from 52.8 °C to 58.4 °C). The A53M substitution inserts a larger hydrophobic methionine inside the hydrophobic pocket formed by Trp50, Tyr81, and Tyr108 (Fig. 6c), stabilizing the active site of Nb4OMT. The E36P-G40E double mutant shifts a glutamate from position 36 to position 40 and thereby preserves the salt bridge interaction with Lys118 while proline capping the alpha helix (Fig. 6d).

To better determine how the A53M substitution affects the substrate recognition of Nb4OMT, GNINA 1.0 was used to dock norbelladine into the crystal structure of Nb4OMT[E36P/G40E/A53M] with SAH and $Ca^{2+}$ already in the active site (based on Fo-Fc electron densities; Supplementary Fig. 14). In the docked structure, the $Ca^{2+}$ ion positions

the catechol moiety of the substrate adjacent to the SAH binding site (Fig. 6e). A similar substrate recruitment by divalent metal ions is found in other, homologous methyltransferases[36,37]. A sulfur-π interaction between the catechol group of norbelladine and Methionine 53 may also restrict the rotation of the catechol group, thereby reducing the cross-methylation of the 3' position and improving specificity.

## Discussion
Herein we report the use of directed evolution and machine learning-guided design for the development of custom microbial biosensors that could be used to monitor substantive improvements in amaryllidaceae alkaloid pathway activity. The RamR transcription factor was evolved to respond to low micromolar levels of the pathway branch-point 4NB. After only four substitutions exquisite specificity emerges for the methylated oxygen moiety in 4NB, with a barely detectable response to the non-methylated precursor norbelladine. The high specificity was also essential for measuring the real-time activity of the plant-derived Nb4OMT enzyme in *E. coli*. Overall, these results highlight the powerful capability of using evolved biosensors for precisely reporting on pathway intermediates while avoiding cross-reactivity with closely related precursor molecules. The RamR protein is now well positioned as an ideal starting point for the generation of biosensors for not only benzylisoquinoline alkaloids, but also for AAs such as galantamine, haemanthamine, lycorine, and their intermediates.

To accelerate our efforts to engineer the Nb4OMT enzyme, we developed a structure-based residual neural network, MutComputeX. Unlike structure prediction models (such as AlphaFold2[26], RosettaFold[38], ESMfold[39], and OmegaFold[40]), or structure-based generative models (such as RFdiffusion[41] and Ig-VAE[42]), to our knowledge, MutcomputeX is the first structure-based model designed to assess sequence substitutions, and that has been explicitly trained to generalize to non-protein atoms, such as nucleic acids and ligands. By

## Table 1 | Kinetic and thermal parameters of the wild-type and mutant Nb4OMTs

| Enzyme | $k_{cat}/K_m$ (µM⁻¹ min⁻¹) | $k_{cat}$ (min⁻¹) | $K_m$ (µM) | $T_m$ (°C) |
|---|---|---|---|---|
| Wild-type Nb4OMT | 1.18 (0.85–1.69) | 73 (63–89) | 62 (37–104) | 52.8 (52.8–52.8) |
| A53M mutant | >2.1 | >190 | <90 | 54.5 (54.3–54.6) |
| E36P/G40E/A53M mutant | >1.4 | >120 | <83 | 58.4 (58.4–58.4) |

Lower and upper bounds for the 95% confidence interval from confidence contour analysis for $k_{cat}/K_m$ and $k_{cat}$ given in parentheses. $K_m$ was calculated by dividing $k_{cat}$ by $k_{cat}/K_m$. Due to the weak substrate inhibition term for the A53M and triple mutation variants, general upper and lower limits on steady state kinetic parameters are reported (see "Methods").

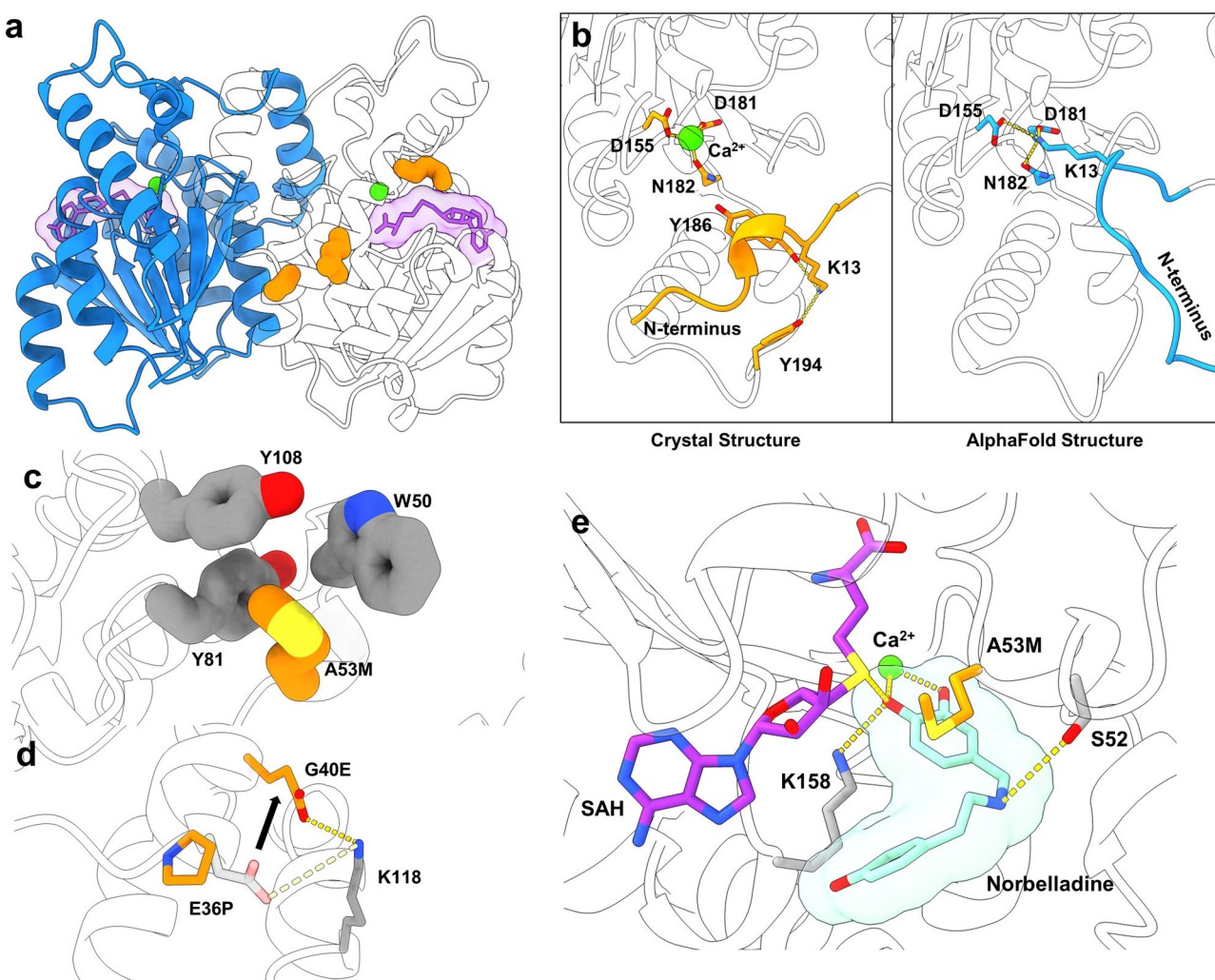

**Fig. 6 | Crystal structure of an engineered norbelladine 4′-O-methyltransferase. a** Global structure of Nb4OMT[E36P/G40E/A53M] solved in 2.4 A resolution. One dimer is colored blue while the other dimer is transparent. **b** Comparison of N-termini of Nb4OMT crystal structure (orange) and wild-type Nb4OMT AlphaFold2 structure (blue). **c** Local context of the A53M mutant residue. **d** Local context of the E36P and G40E mutant residues. Black arrow indicates the shift of glutamate from position 36 to position 40. **e** Active site context of Nb4OMT[E36P/G40E/A53M] in complex with S-Adenosyl-L-homocysteine (SAH) and docked with norbelladine. For (**a**), (**c**), (**d**), and (**e**), the color coding is as follows—calcium ions: green, S-Adenosyl-L-homocysteine: purple, mutant residues: orange, non-mutant residues: gray, docked norbelladine: seafoam green, interactions: yellow dashed lines.

leveraging recent developments in structure prediction (AlphaFold2) and ligand docking (GNINA1.0), we demonstrate that a solved crystal structure is not needed to generate activity-enriched enzyme designs. MutComputeX was trained on ~2.3M microenvironments sampled from over 23,000 protein structures, and facilitated the manual curation of variants with a 60% improvement in product titer, 3-fold lower off-product formation, 5°C higher thermostability, and 2-fold higher catalytic activity. As validation, we show that the ability of MutComputeX to recognize ligand atoms was crucial to predicting the key beneficial A53M mutation in the active site of Nb4OMT.

The solved crystal structure of the engineered Nb4OMT enzyme provides insights into the stabilization afforded by the E36P/G40E substitutions, and the increased activity and regiospecificity afforded by the A53M substitution. Interestingly, the active site of the solved structure differs from AlphaFold2 models, likely due to lack of the metal ions, ligands, and cofactors. While the recently described AlphaFill tool could potentially address this issue for some models, it did not incorporate the norbelladine substrate or provide a more accurate model of Nb4OMT[43] (Supplementary Fig. 15). Together with the 4NB2 sensor and MutComputeX model, the Nb4OMT variant structure should accelerate progress towards further engineering of the AA pathway.

That said, many challenges remain to realizing a commercially viable microbial strain for the fermentation of AAs. First, the central precursors 3,4-dihydroxybenzoate and tyramine must be over-produced in a base strain, likely *Saccharomyces cerevisiae* due to its proven ability to functionally express multiple plant-derived cytochrome proteins[44]. Biosynthetic pathways for the production of 3,4-dihydroxybenzoate and tyramine have already been engineered into *S. cerevisiae*[45,46], with high-throughput screening methods yielding improved tyrosine precursor yields[47]. Second, the early pathway enzymes for the production of norbelladine, norcraugsodine reductase, and norbelladine synthase must be functionally expressed in a microbial strain. So far, the activities of these enzymes have only been demonstrated within an in vitro context[48–50]. Third, the downstream enzymes necessary for the production of advanced AAs must be identified and functionally expressed. With regards to galantamine, the CYP96T1 and CYP96T6 cytochrome proteins have been shown to catalyze the *para-ortho* coupling of 4NB and thus produce the galantamine precursor N-demethylnarwedine[51,52]. Finally, while writing this paper, the remaining NtNMT methyltransferase and NtAKR1 ketone reductase enzymes necessary for complete galantamine biosynthesis have been discovered, unlocking the exciting opportunity to achieve

the biosynthesis of galantamine from simple feedstocks within a microbial host[52].

While the development of strains for microbial AA biosynthesis is a significant endeavor, we believe that the developed 4NB biosensor, ML framework, and enzyme structure presented in this work will significantly accelerate progress towards this goal. Custom biosensor-enabled screens enable rapid collection of phenotype data under a wide variety of experimental conditions, including determining the kinetics of product formation among strain and enzyme variants, values that are nearly impossible to measure using traditional analytical instruments. The importance of machine learning is further highlighted by failed attempts to engineer Nb4OMT using random mutagenesis alone. In the future, microbial semi-synthesis of galantamine and other AAs could provide faster production cycles, a more reliable supply chain, and reduced land and water use compared to traditional plant harvesting methods, and the biosensor-ML hybrid technology stack we have advanced herein should greatly accelerate the engineering of upstream enzymes in the pathway, such as norbelladine synthase and norcraugsodine reductase[48,50].

## Methods

### Strains, plasmids, and media
*E. coli* DH10B (New England Biolabs) was used for all routine cloning and directed evolution. All biosensor systems were characterized in *E. coli* DH10B. LB Miller (LB) medium (BD) was used for routine cloning, fluorescence assays, directed evolution, and orthogonality assays unless specifically noted. LB with 1.5% agar (BD) plates were used for routine cloning and directed evolution. The plasmids described in this work were constructed using Gibson assembly and standard molecular biology techniques. Synthetic genes, obtained as gBlocks, and primers were purchased from IDT. Plasmid designs and sequences are listed in Supplementary Table 7.

### Chemicals
4′-O-methylnorbelladine was purchased from Toronto Research Chemicals (Toronto Research Chemicals. CAT#: H948930). Tyramine (T90344), 3,4-dihydroxybenzaldehyde (37520), dichloromethane (439223), and NaBH$_4$ were purchased from Sigma Aldrich. NMR solvents (d$^6$-DMSO, CD$_3$OD) were purchased from Cambridge isotope laboratories.

### Chemical synthesis and NMR analysis of norbelladine
The aldehyde (3,4-dihydroxybenzaldehyde) (1 mM, 138 mg) and tyramine (1 mM, 137 mg) were dissolved in dichloromethane (5 mL) and converted to the imine in situ compound by stirring for 4 h at room temperature. The imine compound was reduced with NaBH$_4$ (2 mM, 75.6 mg), washed with water and dried to produce crude product. The crude material was then purified by combinatorial flash chromatography to yield norbelladine (10–90% MeCN in H$_2$O, 20 min; 130 mg recovered, beige orange solid, 50% yield), which was confirmed via NMR (Supplementary Fig. 16). NMR spectra were taken on the 500 MHz Bruker prodigy at University of Texas at Austin.

### Chemical transformation
For routine transformations, strains were made competent for chemical transformation. Five milliliters of an overnight culture of DH10B cells was subcultured into 500 mL LB medium and grown at 37 °C and 250 r.p.m. until an optical density of 0.7 was reached (-3 h). Cultures were centrifuged (3500 × *g*, 4 °C, 10 min), and pellets were washed with 70 mL chemical competence buffer (10% glycerol, 100 mM CaCl$_2$) and centrifuged again (3500 × *g*, 4 °C, 10 min). The resulting pellets were resuspended in 20 mL chemical competence buffer. After 30 min on ice, cells were divided into 250-μL aliquots and flash-frozen in liquid nitrogen. Competent cells were stored at −80 °C until use.

### Biosensor response assay
The pReg-RamR and Pramr-GFP plasmids were co-transformed into DH10B cells, which were then plated on LB agar plates containing appropriate antibiotics. Three separate colonies were picked for each transformation and were grown overnight. The following day, 20 μL of each culture was then used to inoculate six separate wells in a 2-mL 96-deep-well plate (Corning, P-DW-20-C-S) sealed with an AeraSeal film (Excel Scientific) containing 900 μL LB medium, one for each test ligand and a solvent control. After 2 h of growth at 37 °C, cultures were induced with 100 μL LB medium containing either 10 μL DMSO or 100 μL LB medium containing the target AA dissolved in 10 μL DMSO. Cultures were grown for an additional 4 h at 37 °C and 250 r.p.m. and subsequently centrifuged (3500 × *g*, 4 °C, 10 min), except in the case of the HPLC comparison assays (Fig. 2f, g), where cultures were grown for an additional 18 h at 37 °C and 250 r.p.m. in order to compare sensor response in fermentation-relevant conditions. Supernatant was removed, and cell pellets were resuspended in 1 mL PBS (137 mM NaCl, 2.7 mM KCl, 10 mM Na$_2$HPO$_4$, 1.8 mM KH$_2$PO$_4$, pH 7.4). One hundred microliters of the cell resuspension for each condition was transferred to a 96-well microtiter plate (Corning, 3904), from which the fluorescence (excitation, 485 nm; emission, 509 nm) and absorbance (600 nm) were measured using the Tecan Infinite M1000 plate reader.

### RamR library design and construction
Three semi-rational libraries were designed, each targeting three inward-facing residues within the RamR ligand-binding pocket (K63, L66, M71; E120, A123, D124; L133, C134, S137) (Supplementary Fig. 1). Libraries were generated using overlap PCR with redundant NNS codons using AccuPrime Pfx (Thermo Fisher, 12344024) and cloned into pReg-RamR. *E. coli* DH10B bearing pSELIS-RamR was transformed with the resulting library. Transformation efficiency always exceeded 10$^6$ for each round of selection, indicating several fold coverage of the library. Transformed cells were grown in LB medium overnight at 37 °C with carbenicillin and chloramphenicol.

### Directed evolution of RamR biosensors
Cell culture (20 μl) bearing the sensor library was seeded into 5 ml fresh LB containing appropriate antibiotics, 100 μg ml$^{-1}$ zeocin (Thermo Fisher, R25001) and 100 μM of norbelladine (for round two) and grown at 37 °C for 7 h. Following incubation, 0.5 μl of culture was diluted into 1 ml LB medium, from which 100 μl was further diluted into 900 μl LB medium. Three hundred microliters of this mixture was then plated across three LB agar plates (100 μL per plate) containing carbenicillin, chloramphenicol and 4NB dissolved in DMSO. Plates were incubated overnight at 37 °C. The following day, the brightest colonies were picked and grown overnight in 1 ml LB medium containing appropriate antibiotics in a 96-deep-well plate sealed with an AeraSeal film at 37 °C. A glycerol stock of cells containing pSELIS-RamR and pReg-RamR encoding the template RamR variant was also inoculated into 5 ml LB for overnight growth.

The following day, 20 μl of each culture was used to inoculate two separate wells in a new 96-deep-well plate containing 900 μl LB medium. Additionally, eight separate wells containing 1 ml LB medium were inoculated with 20 μl of the overnight culture expressing the parental RamR variant. After 2 h of growth at 37 °C, the top half of the 96-well plate was induced with 100 μl LB medium containing 10 μl DMSO, whereas the bottom half of the plate was induced with 100 μl LB medium containing 4NB dissolved in 10 μl DMSO. The concentration of 4NB used for induction is typically the same concentration used in the LB agar plate for screening during that particular round of evolution. Cultures were grown for an additional 4 h at 37 °C and 250 r.p.m. and subsequently centrifuged (3500 × *g*, 4 °C, 10 min). Supernatant was removed, and cell pellets were resuspended in 1 ml PBS. One hundred microliters of the cell resuspension for each condition was transferred to a 96-well microtiter plate, from which the fluorescence (excitation,

485 nm; emission, 509 nm) and absorbance (600 nm) were measured using the Tecan Infinite M1000 plate reader. Clones with the highest signal-to-noise ratio (generally the top 5–10% of the screened clones) were then sequenced and subcloned into a fresh pReg-RamR vector.

For sensor variant validation, the subcloned pReg-RamR vectors expressing the sensor variants were transformed into DH10B cells expressing Pramr-GFP. These cultures were then assayed, as described in the section 'Biosensor response assay', using eight different concentrations of the 4NB. The sensor variant that displayed a combination of low background, a reduced $EC_{50}$ for 4NB and a high signal-to-noise ratio was then used as the template for the next round of evolution.

## Dose–response measurements

Glycerol stocks (20% glycerol) of strains containing the plasmids of interest were inoculated into 1 ml LB medium and grown overnight at 37 °C. Twenty microliters of overnight culture was seeded into 900 µl LB medium containing ampicillin and chloramphenicol in a 2-ml 96-deep-well plate sealed with an AeraSeal film. Following growth at 37 °C and 250 r.p.m. for 2 h, cultures were induced with 100 µl of an LB medium solution containing appropriate antibiotics and the inducer molecule dissolved in 10 µl DMSO. Cultures were grown for an additional 4 h at 37 °C and 250 r.p.m. and subsequently centrifuged (3500 × $g$, 4 °C, 10 min). Supernatant was removed, and cell pellets were resuspended in 1 ml PBS. The cell resuspension (100 µl) for each condition was transferred to a 96-well microtiter plate, from which the fluorescence (excitation, 485 nm; emission, 509 nm) and absorbance (600 nm) were measured using the Tecan Infinite M1000 plate reader.

## Biosensor-linked methyltransferase activity assay

Nb4OMT was expressed with the P150-RBS(riboJ) promoter–RBS on the pReg-RamR plasmid backbone (no regulator present). Cells were co-transformed with both the Nb4OMT plasmid and the 4NB reporter plasmid and plated on an LB agar plate containing appropriate antibiotics. Three individual colonies from each transformation were picked into LB and grown overnight. Resulting cultures were diluted 50-fold into 1 mL LB medium containing the indicated concentration of norbelladine in a 96-deep-well plate and were grown at the indicated temperature for 24 h. Subsequently, the fluorescence of cultures was measured in the same manner as previously described in 'Dose–response measurements' above.

## Protein-cofactor-substrate complex generation with Alpha-Fold2 and GNINA1.0

Nb4OMT wild-type sequence (Uniprot id: A0A077EWA5) was run through the AlphaFold2-multimer as a homodimer using the publicly available collab notebook. This resulted in a computational structure with a pLDDT of 0.955 and a pTM of 0.94. The initial coordinates for the SAH cofactor were transplanted onto the AlphaFold2 structure from the 1SUI PDB structure and then optimized with GNINA1.0's –local_only and –minimize flags. Norbelladine's initial 3D coordinates were obtained from the PubChem database (id: 416247) and docked into the active site of the A protomer. To dock norbelladine, we generated a bounding box for the GNINA docking procedure by finding the largest 3D box from the atomic coordinates of the following residues: L10, W50, S52, A53, D155, D157, K158, W185, Y186, A204. GNINA was run several times with different seeds and all docked poses were manually screened for known mechanistic insight (orientation with K. The docked pose that best satisfied the mechanistic insight and received a high GNINA docking score was then minimized with the –local_only and –minimize flags. The docking results from GNINA for SAH and NB are shown in Supplementary Table 1.

## Building MutComputeX

**Structure File pre-processing.** To generate voxelized matrices of microenvironments that span between protein:non-protein atoms, experimental CIF files were pre-processed with (1) ChimeraX to add hydrogen atoms to the proteins, nucleic acids, and organic ligands; (2) ChargeFW2 to add polarized charges that bridge protein: non-protein interfaces; and (3) FreeSASA to add solvent accessible surface area values that take into account protein:non-protein interactions. CIF read and write functionality for ChargeFW2[53] and FreeSASA[54] were implemented and merged to both open-sourced libraries.

**Voxelized matrix generation.** To generate a voxelized molecular representation of a microenvironment, a 20 Å cube of atoms was filtered from the structure centered on the Calpha and oriented with respect to the backbone where the side chain was along the +z axis. All atoms in the center residue are then removed prior to insertion into a voxelized grid with 1 Å resolution. Each atom is placed into a corresponding element channel except halogen atoms (which are placed into a multi-atom channel that consist of F, Cl, Br, I), resulting in the following atomic channels: C, H, O, N, S, P, Halogens. Each atom's partial charge and SASA value are placed into the partial charge and SASA channels, respectively. For all 9 channels, atom values are gaussian blurred according to their Van-der-Waals radii. The P and Halogen channels were added to the original MutCompute framework in order to generalize to ligands and nucleic acids.

**Dataset generation.** A dataset of 50% sequence similar protein chains with at least a 3.0 Å resolution was downloaded in November 2021 from the RCSB. This provided us with 22584 protein sequences from 21613 PDB entries. To generate microenvironment datasets, for each protein chain we prioritized residues that were within 5 Å of a non-protein entity, via the GEMMI[55] library ContactSearch functionality, and then randomly backfilled until 200 residues or half of the protein sequence was sampled. A total of 2,569,256 microenvironments were sampled from 22,584 protein sequences and split 90:10 to generate our training and test set splits for interfaces and non-interface residues are shown in Supplementary Table 5.

**Model training.** The 3D residual neural network was built in Tensorflow 2.7. The architecture is provided in Supplementary Fig. 7. Each model run was parallelized over 4 AMD Radeon Instinct MI50s with a batch size of 200. Models were trained for up to 8 epochs where each epoch was saved as a checkpoint with a variety of hyperparameters. We used a scheduled learning rate that began at 0.001 and had an exponential decay constant of either 0.3 or 0.5 and an adaptive learning rate that would lower the learning rate by 0.25 if the training accuracy did not improve by 0.1% after either 30 K, 50 K, and 60 K training instances. Weights were updated with the Adam optimizer and all convolutional layers had weight decay regularization of 0.001.

**Model benchmarking.** To ensure our datasets were enabling the 3DResNet models to generalize across protein:non-protein interfaces, we monitored the overall wild-type accuracy and wild-type accuracy for residues at DNA, RNA, and ligand interfaces on our test set. To select models to ensemble and generate engineering predictions we generated zero shot-predictions for all mutational data in FireProtDB and chose the models that had the highest correlation with the single point mutation ΔTM experimental data. The zero-shot predictions were generated by taking the prediction assigned to the wild type and mutant amino acid from FireProtDB and taking the log odds where a positive log odd means a stabilizing prediction and a negative log odd means a destabilizing prediction. The ensembled model had a Pearson and Spearman correlation coefficients of 0.367 and 0.425 with the 2719 single point mutations with ΔTM experimental data in FireProtDB and a Pearson and Spearman correlation coefficients of −0.407 and −0.457 with the 4889 single point mutations with ΔΔG experimental data in FireProtDB. Correlation coefficients for the independent models can be found in Supplementary Table 6.

## Mutational designs

Mutations were designed with two goals: stabilizing the protein away from the active site and investigating point mutations where predictions differed between the docked and apo protein structures. With these objectives, we sorted residues based on the log odds between the predicted and wild-type amino acids. For the stability objective, predictions that recapitulate known chemical phenomena such as salt bridges, hydrogen bonding, proline capping were prioritized. An in-depth discussion of the mutation curation process is described in Supplementary Discussion 1.

## High-performance liquid chromatography analysis

Assay samples were filtered using a 0.2-μm PTFE syringe filter prior to running the HPLC. The measurement of Norbelladine and 4′-O-methylnorbelladine was performed using a Vanquish HPLC system (Thermo Fisher Scientific) equipped with a BDS Hypersil TM C18 (3.0 × 150 mm², 3 μm) (Thermo Fisher Scientific) with detection wavelength 277 nm. The mobile phase consisted of 0.1% formic acid in water or 0.1% formic acid in acetonitrile over the course of 28 min under the following conditions: 10% organic (vol/vol) for 2 min, 10 to 30% organic (vol/vol) for 13 min, 30 to 90% organic (vol/vol) for 0.1 min, 90% organic (vol/vol) for 4.9 min, 90 to 10% organic (vol/vol) for 1 min, and 10% organic (vol/vol) for 7 min. The flow rate was fixed at 0.8 ml min⁻¹. A standard curve for norbelladine was prepared using synthesized norbelladine (see 'Chemical synthesis and NMR analysis of norbelladine'). A standard curve for 4′-O-methylnorbelladine was prepared using commercially available 4′-O-methylnorbelladine.

Reactions for kinetics measurements were performed in triplicate for all enzyme variants. For each variant, 1.5 ml reactions containing 3.5 nM of enzyme, 500 μM SAM, 2 mM CaCl₂, and 15.625, 31.25, 62.5, 125, 250, or 500 μM norbelladine in PBS pH 7.5 were incubated at 37 °C for 4 h. Every hour a 200 μl aliquot of each reaction was quenched by pipetting it into a 1.5 ml microcentrifuge tube with 20 μl of 2 M HCl. The concentration of 4′-O-methylnorbelladine was then determined using HPLC as described.

## Liquid chromatography–mass spectrometry

Cells containing the plasmid expressing each Nb4OMT variant with the P150-RBS(RiboJ) promoter were transformed and plated onto an LB agar plate containing appropriate antibiotics. The following day, three colonies from each plate ($n = 3$) were cultured overnight in LB and subsequently diluted 50-fold into 1 ml LB containing 1 mM norbelladine. These cultures were grown for 24 h at 37 °C and centrifuged at 16,000 × $g$ for 1 min, and the resulting supernatant was filtered using a 0.2-μm filter.

Samples were analyzed using an Agilent 6530 Q-TOF LC–MS with a dual Agilent Jet Stream electrospray ionization source in positive mode. Chromatographic separations were obtained under gradient conditions by injecting 10 μl onto an Agilent RRHD Eclipse Plus C18 column (50 × 2.1 mm, 1.8-μm particle size) with an Agilent ZORBAX Eclipse Plus C18 narrow-bore guard column (12.5 × 2.1 mm, 5-μm particle size) on an Agilent 1260 Infinity II liquid chromatography system. The mobile phase consisted of eluent A (water with 0.1% formic acid) and eluent B (acetonitrile). The gradient was as follows: Hold 95% A/5% B from 0 to 2 min (0.7 ml min⁻¹), 80% A/20% B from 2 to 15 min (0.7 ml min⁻¹), 70% A/5% B from 15 to 18 min (0.7 ml min⁻¹). The sample tray and column compartment were set to 7 °C and 30 °C, respectively. The fragmentor was set to 100 V. Q-TOF data were processed using the Agilent MassHunter Qualitative Analysis software (Version 10.0). Both products and the residual substrate of the wild-type reactions were identified with MS/MS with a collision cell energy of 5 V. To create the chromatograms (shown in Fig. 4C and Supplementary Fig. 6), signal counts from the EIC within a window ±0.05 min relative to the retention time of the substrate and products were extracted for each scan ($m/z$ ratios 260.1281 and 274.1438).

## Enzyme kinetics calculations

Kinetic data were fit in KinTek Explorer simulation and data fitting software v11[56]. The following minimal model was used as an input. Each line represents a step in the model and the forward reaction goes from left to right while the reverse reaction goes from right to left as written.

(1) E + S = ES
(2) ES = EP
(3) EP = E + P
(4) S = P2

Starting concentrations were entered into the software just as the reactions were performed:

3.5 nM enzyme and 15.625, 31.25, 62.5, 125, 250, and 500 M substrate. The output observable was defined as EP + P. Substrate oxidation was modeled in step (4) as irreversible with a best-fit value from globally fitting data from all variants to derive $k_4 = 0.00547$ min⁻¹. To get $k_{cat}/K_m$ and $k_{cat}$: $k_{-1}$, $k_{-2}$, and $k_{-3}$ were locked at 0 min⁻¹ (irreversible reactions). $k_{+3}$ was locked at 10,000 min⁻¹ as to not limit the rate of turnover. $k_{+1}$ and $k_{+2}$ were used as variable parameters in the fitting. Under these conditions, $k_{+2} = k_{cat}$ and $k_{+1} = k_{cat}/K_m$. For estimates of 95% confidence intervals on kinetic parameters, confidence contour analysis was used with the FitSpace function in KinTek Explorer[57]. Confidence contour plots are calculated by systematically varying a single rate constant and holding it fixed at a particular value while refitting the data, allowing other rate constants to float. The goodness of fit was scored by the resulting $\chi^2$ value. The confidence interval is defined based on a threshold in $\chi^2$ calculated from the F-distribution based on the number of data points and number of variable parameters to give the 95% confidence limits. For the data given in Supplementary Fig. 12, this threshold was 0.85 to estimate the upper and lower limits for each parameter. While the model described above is the simplest model that could describe the data and gave reasonable estimates for $k_{cat}$ and $k_{cat}/K_m$, there was evidence for substrate inhibition at the highest norbelladine concentration for the two variants (A53M and triple mutant) that this model did not account for. We then fit the data for these two variants to the model shown below, accounting for substrate inhibition.

(1) E + S = ES
(2) ES = EP
(3) EP = E + P
(4) E + S = SE
(5) SE + S = SES
(6) ES + S = SES
(7) S = P2

As before, $k_{+1}$ was allowed to float in the fitting to give $k_{cat}/K_m$, and $k_{-1}$ was locked at 0 min⁻¹. $k_{+2}$ was allowed to float in the fitting to give $k_{cat}$, and $k_{-2}$ was locked at 0 min⁻¹. $k_{+3}$ was locked at 10,000 min⁻¹, and $k_{-3}$ was locked at 0 min⁻¹. $k_{+4}$ and $k_{+6}$ were locked at 100 μM⁻¹ min⁻¹, and $k_{-4}$ and $k_{-6}$ were allowed to float in the fitting as linked parameters. $k_{+5}$ was linked to $k_{+1}$ and $k_{-5}$ was locked at 0 min⁻¹. $k_{-7}$ was locked at 0 min⁻¹, and $k_{+7}$ was locked at 0.00547 min⁻¹. With limited inhibition at the highest substrate concentrations tested, confidence contour analysis showed that only lower limits on $k_{cat}$, $k_{cat}/K_m$, and substrate inhibition could be obtained from the analysis, and these limits are reported in Table 1.

## Protein expression and purification

For bacterial overexpression of Nb4OMT wild type and its variants (A53M and E36P + G40E + A53M), *E. coli* BL21 (DE3) was used as the expression host and its competent cell was transformed with the corresponding constructed plasmids. A single colony of an *E. coli* BL21 (DE3) strain harboring one of the constructed plasmids was inoculated into 2 mL of Luria Bertani broth (LB) medium with 100 μg/mL ampicillin and grown overnight at 37 °C/225 rpm. The overnight-grown culture (using 1 mL) was scaled up into a 500-mL autoinduction media at 37 °C/225 rpm. Protein expression was

automatically induced and cells were cultured for 24 h at 25 °C/225 rpm. The induced cell culture was harvested by centrifugation at 4000 × g and 4 °C for 20 min. Cell pellets were then resuspended in 200 mL of lysis buffer (50 mM TRIS pH 8.0, 500 mM NaCl, 20 mM Imidazole, 10% Glycerol, 10 mM β-mercaptoethanol, and 0.1% Triton-X). Cells were lysed by sonication and the resulting cell lysate was centrifuged at 15,000 × g and 4 °C for 20 min to obtain the supernatant that contains soluble proteins. The supernatant was equilibrated with HisPur™ Ni-NTA Resin (Thermo Fisher Scientific, Waltham, MA) and washed with 10x bed volumes of wash buffer (50 mM TRIS pH 8.0, 500 mM NaCl, 20 mM Imidazole, 10% Glycerol, 10 mM β-mercaptoethanol). Then protein was eluted by using a 10 mL elution buffer (50 mM TRIS pH 8.0, 500 mM NaCl, 250 mM Imidazole, 10% Glycerol, 10 mM β-mercaptoethanol). The eluate was dialyzed with 3 C protease added to the dialysis cassette, into the appropriate buffer (20 mM TRIS pH 7.5, 100 mM NaCl, 10 mM β-mercaptoethanol) followed by size-exclusion fast protein liquid chromatography. All Nb4OMT variants were stored in 20 mM Tris (pH 7.5), 100 mM NaCl and 10 mM β-mercaptoethanol.

### Protein X-ray crystallography
To identify crystallization conditions of the Nb4OMT variant with triple mutations (E36P + G40E + A53M), 20 mg/ml purified enzyme samples were directly used in sparse matrix screening. Rod-shaped crystals formed after incubating screening plates at room temperature for 3 days. A crystallization condition with the best crystal morphology (0.1 M Calcium Acetate, 0.1 M MES pH6.5, and 20% PEG3350) was chosen and further optimized by manually setting sitting-drop vapor diffusion experiments by varying pH and precipitant concentration, resulting diffraction-quality single crystals in 0.1 M Calcium Acetate, 0.1 M MES pH 7.0, and 26% PEG3350.

Individual Nb4OMT variant (E36P + G40E + A53M) crystals were flash-frozen directly in liquid nitrogen after brief incubation with a reservoir solution supplemented with 30% (v/v) glycerol. X-ray diffraction data were collected at BL 8.2.2 in ALS (Berkeley, CA). X-ray diffraction data were processed to 2.4 Å using HKL2000. In Phenix software, phases were obtained by molecular replacement using an AlphaFold2 model of Nb4OMT as the initial search model. The molecular replacement solution was iteratively built and refined using Coot and Phenix refine package. The quality of the final refined structures was evaluated by MolProbity. The final statistics for data collection and structure determination are shown in Supplementary Table 4.

### Differential scanning fluorimetry
Purified Nb4OMT variants in the concentration of 5 μM were prepared in 96-well low-profile PCR plates (ABgene, Thermo Scientific). 10X SYPRO® Orange (Molecular Probes) was added into each well and mixed prior to measurement in an RT-PCR machine (LightCycler 480, Roche). The protein melting experiments were carried out with a continuous temperature acquisition mode using 10 acquisitions per 1 °C in each cycle from 20 °C to 95 °C. The melting curves of the Nb4OMT variants were monophasic and $T_m$ values were derived using Boltzmann equation.

### Statistical analysis and reproducibility
All data in the text are displayed as mean ± S.D. unless specifically indicated. Bar graphs, fluorescence and growth curves, dose–response functions were all plotted in Python 3.6.9 using Matplotlib. Dose–response curves and $EC_{50}$ values were estimated by fitting to the Hill equation $y = d + (a − d)x^b(c^b + x^b)^{−1}$ (where y = output signal, b = Hill coefficient, x = ligand concentration, d = background signal, a = maximum signal and c = $EC_{50}$), with the scipy.optimize.curve_fit library in Python.

### Reporting summary
Further information on research design is available in the Nature Portfolio Reporting Summary linked to this article.

## Data availability
Protein sequence information was retrieved from the NCBI database: RamR, 3VVX_A; Nb4OMT, A0A077EWA5. Plasmid sequences relevant to this study can be found in Supplementary Table 7 and have been deposited in Addgene (216231, 216232). The Alfalfa caffeoyl coenzyme A 3′-O-methyltransferase (PDB: 1SUI) was used to assist with docking. Coordinates for the complex structure of Nb4OMT^E36P/G40E/A53M with S-adenosyl-L-homocysteine (SAH) has been deposited in the Protein Data Bank (PDB) as 8UKE. Source data are provided with this paper.

## Code availability
Code used to generate bar plots and dose–response functions presented in this text is accessible at https://github.com/simonsnitz/plotting[58]. The MutComputeX model as well as the input data of the norbelladine-4O-methyltransferase are available at https://github.com/danny305/MutComputeX[59].

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

## Acknowledgements

Funding from National Institute of Standards and Technology (70NANB21H100 to A.D.E.), and the Air Force Office of Scientific Research (FA9550-14-1-0089 to A.D.E.) is acknowledged. This work was

partially supported by grants from the National Institutes of Health (R35GM148356 to Y.Z. and R01EB026533 to A.D.E.). We would like to thank NSF AI Institute for Foundations of Machine Learning (IFML) for their support and AMD for the donation of hardware and support resources from its HPC fund. Crystallographic data collections were conducted at Advanced Light Source (BCSB - BL 8.2.2), Department of Energy scientific user facility at Lawrence Berkeley National Laboratory.

## Author contributions

S.d'O. conceived the project, designed experiments, and performed biosensor evolution and characterization with support from M.W.S. D.J.D. developed MutComputeX with support from J.M.L. and enzyme designs were generated by D.J.D. S.d'O. performed enzyme engineering with support from H.D. D.J.A. performed HPLC measurements with support from S.d'O. T.L.D. performed enzyme kinetic assays and analysis, with support from D.J.A. M.B.M. synthesized norbelladine. J.R.H. conducted LC/MS analysis. W.K. purified protein and conducted X-ray crystallography with support from Y.J.Z. The manuscript was written by S.d'O. with support from W.K., D.J.D., A.D.E., Y.J.Z., and H.S.A. S.d'O. and D.J.D. supervised all aspects of the study.

## Competing interests

The authors declare the following competing interests. Patent applicant: The Board of Trustees of the University of Texas at Austin. Name of inventor(s): S.D., D.J.D., and A.D.E. Title: METHODS AND COMPOSITIONS RELATED TO MODIFIED METHYLTRANSFERASES AND ENGINEERED BIOSENSORS. Application number: 18/436,635. Status of application: Pending. S.D. has financial relationships with Retna Bio LLC, and D.J.D. has financial relationships with Intelligent Proteins LLC. All other authors declare no competing interests.
