## [Peer Review File · Nature Communications]

REVIEWER COMMENTS

Reviewer #1 (Remarks to the Author):

In the presented study the authors show-case a follow-up study based on their previous work on RamR as an evolvable biosensor-scaffold useful for high-throughput screening of relevance to biocatalysis optimization. Specifically, in this study they demonstrate the RamR versatility by finding RamR-variants able to detect Amaryllidaceae alkaloids in particular towards the common precursor 4NB. The authors used their previously developed toggle-selection regime SELIS to develop a biosensor for 4NB with a detection limit of 2.5 μM and 80-fold selectivity for 4NB over its precursor NB. With a biosensor at hand the authors aimed to use it for screening of Nb4OMT enzyme variants designed based on machine learning to accelerate the discovery of enzymes and pathways with improved stability and activity.

To do so they rebuilt the data engineering pipelines from MutComputeX to accept cofactors and ligands for Nb4OMT, demonstrating superior feature extraction capabilities compared to previous model. Ultimately the authors demonstrate the identification of Nb4OMT variant enzymes with up to 60% improvement in product formation compared to wild-type Nb4OMT, while the level of substrate norbelladine decreased 17-fold, while off-product decreased by approx. 3-fold.

The study convincingly makes use of a very powerful approach combining directed evolution and machine learning for metabolic engineering/biotech applications, and with demonstrated success both in terms of biosensor engineering and application. Furthermore, being a common precursor to a multitude of AAs, the 4NB biosensor should be useful for further pathway discovery and 4NB optimization efforts. This also goes for the Nb4OMT variants.

However, the general approach to directed evolution of biosensors, and the ML-/AlphaFold-guided enzyme variant library design is not novel. Neither is the application of the biosensor used for functionalizing upstream 4NB enzymes, alike norcraugsodine reductase, and norbelladine synthase, never previously expressed in microbes. Likewise, no generally applicable mechanistic understanding of the engineered MTase is provided. These points all limit the scope of the work presented beyond the impactful previous studies from this team. According to this reviewer, this precludes publishing the study in this journal. This said the optimization of MutComputeX is very interesting, and so are the novel RamR and enzyme variant, and the study should be suitable for a more specialised journal focusing on method development for metabolic/enzyme engineering or computational biology.

Below a few points are listed which hopefully can help the authors further improve their interesting study.

Major:

1. In the first Result paragraph it states that norbelladine was docked on to RamR, yet both legends in Fig 1c and Suppl. Fig. 1 states 4NB. In the second Result paragraph it (correctly?) states 4NB-based docking to inform library design. Please explain or correct.

2. In relation to Fig. 2 it is unclear to this reviewer what difference there is between panel b) and f) when it comes to RFU/OD. In both panels biosensor 4NB2.1 is used yet panel b) has a signal of $0-5 \times 10^4$, while panel f) shows $0-6 \times 10^5$ RFU/OD. Please explain the shift in signal. In relation to these two plots it would also be needed to further elaborate on the operational range of the 4NB2.1 biosensor. In panel b) it seems to saturate at approx 100 μ M 4NB, while in panel f) it is evident that a (significant?) difference can be observed between 100 and 250 μ M.

3. On the note "...it failed to enhance Nb4OMT activity", from the epPCR variant library screen, could the authors provide speculations or data indicating why the phenotypes from colonies on plates to population/single cell fluorescence from liquid cultures did not correlate? Plate screens are very powerful, and dirt cheap. Readers would benefit from the authors' considerations here.

4. Since epPCR library size of Nb4OMT is not estimated/indicated and that no variants with enhanced activity was identified from the plate assay, it is unclear to this reviewer why the authors move on to adopt machine learning based on structure-guided engineering. The plate-screen is claimed to yield different fluorescence intensities (even though this data is not presented), and if this was the case why would colonies from this assay not form relevant training data for ML?

5. It is an over-statement to say that a structure-based CNN-based approach was pursued when no Nb4OMT structure is available to support the training data. Authors should rather state that CNN was considered but aborted due to lack of structural data.

Minor:

L3 in Intro. Please insert what AA stands for - Amaryllidoideae alkaloids?

Legend for Suppl. Fig. 2 needs updating on numbering of subpanels. Also, no indication of replicates is provided in the figure legend. Please update this, incl indication of error bars as well as statistical testing if applied.

Reviewer #2 (Remarks to the Author):

This manuscript is focused on developing better ways of biocatalytic production of 4-O-Methylnorbelladine (4NB) from norbelladine (Nb) via directed evolution and machine learning-based protein engineering. The first part covers two rounds of directed evolution of the RamR repressor for detecting 4NB, based on three hotspots identified by docking 4NB and the SELIS method published earlier. These two rounds led to a significantly more sensitive (4NB) and specific (4NB vs Nb) variant, which was then coupled with Nb 4-O-methyltransferase (Nb4OMT) in *E. coli* to create an in vivo assay for activity measurement, used for directed evolution of Nb4OMT in the second part.

For evolving Nb4OMT, the authors first applied random mutagenesis, according to the authors, with little success. Then they modelled the Nb4OMT structure with AlphaFold2, docked Nb and a cofactor, and applied a modification of MutCompute (retrained on protein-ligand complexes) to suggest substitutions. After experimentally testing 22 mutants manually selected based on the MutCompute output, the authors ended up with several improved designs.

In general, the study covers an impressive workflow consisting of biosensor development, self-supervised learning, and protein engineering. The results are novel in each part, and the study will definitely benefit biochemists and protein engineers. However, while the work is exciting, the reporting sometimes lacks clarity, and some of the claims seem too strong for the evidence provided in the text. I outline my reservations below.

Directed evolution of the 4NB sensor:

- After the first round of the directed evolution targeting sensitivity for 4NB, the authors chose 4NB1.2 as a starting point for the second round, which targeted counter-selection against Nb. But based on the data shown in Figure S2, 4NB1.1 and 4NB1.3 had higher sensitivity and seemed to be better candidates for the second round. Moreover, 4NB1.2 already showed close-to-zero sensitivity for Nb, so evolving it for the counter-selection against Nb seems counterintuitive. Could the authors please comment on why they preferred 4NB1.2?

- The authors highlight the superiority of the 4NB sensor over HPLC based on the lower dynamic ranges for the concentration of 4NB (2.5 μ M -- 250 μ M vs. 25 μ M -- 1000 μ M) and refer to Figures S2c, d and Figure 2f as evidence. However, those figures are not convincing in this respect as they do not seem to cover the low concentration values in detail. What was the 4NB concentration for the measurements reported in Figures S2 b,d?

Directed evolution of Nb4OMT:

- The authors mention that the new self-supervised 3DResNet achieved improved wildtype prediction accuracy 80% compared to 69% for the previously published version. However, after ensemble-based fine-tuning using the data from FireProtDB, the protein-ligand interface wildtype accuracy dropped to 66%. Why not use the original model then?
- It is striking that only one mutation showed a consistent increase in the signal at low temperatures because in other studies using MutCompute, the success rate was much higher (8-9 out of 10). Can it be that the AlphaFold2-generated structure is unreliable?
- “MutComputeX ... predicted functional variants of the Nb4OMT enzyme with 60% improvement in product titer, 17-fold reduced remnant substrate, and 3-fold lower off-product formation” should be rephrased as “MutComputeX... allowed constructing several variants with 60% improvement ...”.
- As I understand, the mutant selection for experimental validation involved significant manual curation of MutComputeX output. This should be mentioned explicitly in the results and discussion, and the selection deserves more details than just a short sentence at the end of Mutational Designs. Also, the statement “We believe that the approach presented in this work, synergizing custom biosensor-enabled screens with self-supervised machine learning-guided protein design, will fundamentally accelerate the pace of strain and enzyme engineering as a whole” is an exaggeration as the study does not demonstrate the generalisability of the protocols to other systems. The manual curation step implies that the approach is quite case-study-specific.
- I could not find any information on the numbering of OMT variants in figures, e.g., see Figure 4b. Could the authors please provide a table with the exact substitutions introduced to each variant?

Training of MutComputeX:

- How many atom channels were introduced to the original architecture because of the ligands?
- “This provided us with X protein sequences from Y PDB entries.” What are X and Y?
- How was the prioritization of residues within 5Å of non-protein entities implemented?
- Did the authors consider calculating the correlations with FireProtDB data for the scores returned by the original MutCompute method?

Minor comments:

- In many graphs, the authors mention that error bars represent the S.E.M +/- the mean, but they look more like standard deviations to me. In fact, I would report the latter for triplicates.
- In the first paragraph of the results, the authors mentioned the docking of Nb at RamR. Did they mean 4NB? (based on Figure 1C and the next paragraph).
- The authors mention that random error-prone PCR failed to enhance Nb4OMT activity. It will be valuable to see what activity (fluorescence) distribution the authors observed in this step to have a better idea of the baseline.
- Figure S5 is somewhat confusing. The text under the x-axis (Fed substrate μM) seems to refer to the legend rather than the axis. Could the authors comment on why the concentrations of the substrate and product do not sum up to the input concentrations?
- Many abbreviations and names of the tools are spelled differently throughout the manuscript and should be unified (AlphaFold2 – Alpha fold, Nb4OMT – OMT - 4OMT, etc).

Reviewer #3 (Remarks to the Author):

Amaryllidaceae alkaloids are a special group of alkaloids presented in the plants of the family Amaryllidaceae. Some of them such as galanthamine are well known for its therapeutic application for the patients with Alzheimer's disease. Generally speaking, it is very interesting and challenging to reconstitute a whole biosynthetic pathway in *E. coli* to produce the specific plant natural product of interest. In the present study, the authors claimed a synthetic microbial sensing and biosynthesis of Amaryllidaceae alkaloids. Directed evolution of RamR, a starting point for identifying biosensors for a variety of benzylisoquinoline alkaloids, led to a specific biosensor for 4'-O-methyl-norbelladine. However, directed evolution of the O-methyltransferase Nb4OMT using the error-prone PCR for saturated mutagenesis did not enhance its catalytic activity. Thus the authors developed an AI technology model MutComputeX to generate catalytic activity-enhanced mutant for Nb4OMT. The experimental results showed a 60% improvement of the yield of 4'-O-methyl-norbelladine, 17-fold reduction of remained substrate, and 3-fold lower of the production of 3'-O-methyl-norbelladine. All results are very interesting. However, the novelty of the whole study is very limited.

4'-O-Methyl-norbelladine has been thought to be the branch point of the Amaryllidaceae alkaloids biosynthesis. It is very important to produce 4'-O-methyl-norbelladine with high yield in *E. coli*. However, it is not difficult to chemically synthesize 4'-O-methyl-norbelladine with high yield. In the biosynthesis of Amaryllidaceae alkaloids, the O-methylation is not the rate-limiting reaction step. Additionally, it is not the entire pathway for 4'-O-methyl-norbelladine but only one enzyme, i.e., Nb4OMT, was investigated in the present work.

Directed evolution is a useful method to enhance the catalytic activity of enzyme. RamR had been applied in identification of biosensor for some alkaloids. The directed evolution of RamR led to a specific biosensor for 4'-O-methyl-norbelladine. However, it failed to enhance the catalytic activity of Nb4OMT. In the present study the authors employed error-prone PCR to mutant Nb4OMT. More rational directed evolution methods may give good results.

MutComputeX model was developed to guide the evolution of Nb4OMT. The results are kind of good. Especially, the A53M mutant is the better one. Is there some reasonable explanation for this mutation?

Point-by-point response to Reviewer comments

Key additions to the manuscript include the following:

1. A new figure (Figure 4) covering the machine learning pipeline, to highlight the novelty of our MutcomputeX model.
2. A new complex structure of an engineered norbelladine 4-O'Methyltransferase variant in complex with SAH in the active site at 2.4 Å, which is described in a new figure (Figure 6) that elucidates the structural mechanism underlying key beneficial mutations.
3. A new table (Table 1) containing kinetic and thermodynamic parameters of the wild-type and engineered methyltransferase enzymes.
4. A new Supplementary Figure (Figure S11) comparing MutComputeX predictions when the substrate and cofactor ligand atoms are and are not accounted for.
5. A new Supplementary Figure (Figure S12) illustrating the datasets and confidence contours used to calculate enzyme kinetic parameters
6. A new Supplementary Figure (Figure S15) describing the Nb4OMT model created using AlphaFill.
7. A new Supplementary Table (Table S1) comparing 4'-Omethylnorbelladine measurements with HPLC and biosensor methods using a wide range of ligand concentrations.
8. A new Supplementary Table (Table S3) describing the alias and genotypes of combinatorial Nb4OMT mutants.
9. One new Supplementary Table (Table S4) and two new Supplementary Figures (Figure S13 & S14) describing crystallography metrics, protein chromatography profiles, and the omit F_oF_c map of the solved crystal structure, respectively.
10. A new Supplementary Discussion (Discussion S1), which includes an in-depth description of the manual curation process for Nb4OMT variant design.

Major Point #1: Clarify novelty

We believe that the largest sources of novelty within this paper come from (1), the development and application of the first ligand-aware, structure-based AI model for enzyme engineering; (2) the development and application of the first protein biosensor for any amaryllidaceae alkaloid; and (3) the first crystal structure of an amaryllidaceae alkaloid methyltransferase.

- (1) The development and application of the first ligand-aware, structure-based AI model for enzyme engineering.

To clarify the substantial advance represented by our structure-based, AI model, we have created a separate figure (**Figure 4**) illustrating the substantial improvements made to our previously developed AI model, MutCompute, as well as added a new accompanying section entitled “Developing a machine learning pipeline for structure-based enzyme engineering.”

Figure 4. The MutComputeX pipeline.

(a) The A53 microenvironment inputs for MutCompute (top), which does not include non-protein atoms, and for MutComputeX (bottom), which includes both ligand and cofactor atoms (PDB: 8UKE). (b) The microenvironment is voxelized into seven elemental and two physical channels. All halogen atoms are combined into a single channel. (c) An overview of MutComputeX residual neural network architectures. A more detailed architecture diagram is provided in **Supplementary Figure 7**. (d) Workflow using MutComputeX for enzyme engineering. In the A53 masked microenvironment that is shown, the orange spheres represents the masked alanine, the norbelladine ligand is shown in aqua, protein residues are shown in gray, and S-adenosyl-homocysteine (SAH) is shown in purple. (p. 21)

We have also made text edits to highlight the novelty of MutComputeX and compare it to existing work in the **Results and Discussion**:

Results: “Developing a machine learning pipeline for structure-based enzyme engineering”

The original data engineering pipelines established for MutCompute restricted its training to microenvironments with atoms belonging to the 20 amino acids, and therefore MutCompute was unable to provide contextualized predictions in microenvironments that possessed atoms from cofactors, ligands, or nucleic acids^{29,31}. To create designs that could be generalized to protein-ligand interfaces, we have developed MutComputeX: the first structured-based neural network designed to generalize to microenvironments at protein-ligand, -nucleotide, and -protein interfaces (**Figure 4a**). To develop MutComputeX, we first rebuilt the data engineering pipelines to enable training on heterogenous microenvironments (**Figure 4b**). New atomic channels for phosphorus and grouped halogens were added to the input representation (see Methods). New training and testing datasets were curated that included sampling ~256,000 protein-ligand interface microenvironments (see Methods). Finally, a novel residual convolutional architecture was developed to improve feature extraction capabilities and in turn the predictive power of the model^{32,33} (**Figure 4c**, Supplementary Figure 7) (**p. 9**)

Discussion

To accelerate our efforts to engineer the Nb4OMT enzyme, we developed a structure-based residual neural network, MutComputeX. Unlike protein language models (such as ESM-1b³⁸, ESM2³⁹, and ProtTrans⁴⁰), or other structure-based models (such as MutCompute²⁹, ProteinMPNN⁴¹, and ESM-IF⁴²), MutcomputeX is the first model that has been explicitly trained to generalize to non-protein atoms, such as nucleic acids and ligands. By leveraging recent developments in structure prediction (AlphaFold2) and ligand docking (GNINA1.0), we demonstrate that a solved crystal structure is not needed to generate activity-enriched enzyme designs. MutComputeX was trained on ~2.3M microenvironments sampled from over 23,000 protein structures, and facilitated the manual curation of variants with a 60% improvement in product titer, 3-fold lower off-product formation, 5°C higher thermostability, and 4-fold higher catalytic activity. As validation, we show that the ability of MutComputeX to recognize ligand atoms was crucial to predicting the key beneficial A53M mutation in the active site of Nb4OMT. (**p. 14**)

We have also included an analysis that shows that the advancements made to MutComputeX - the incorporation of protein:non-protein atoms - was essential to predicting the key beneficial A53M mutation:

Results: “Characterization of improved norbelladine methyltransferase variants”

Interestingly, we found that the beneficial A53M substitution was only predicted by MutComputeX when the Nb4OMT structure model was docked with SAH and norbelladine; in contrast, A53R was predicted when docking was not performed, a substitution that reduced activity under all tested conditions (**Supplementary Figure 8**, **Supplementary Figure 11**). These results clearly demonstrate that the incorporation of ligand atoms to the microenvironment greatly improves MutComputeX’s ability to engineer the active site of enzymes. (**p. 11**)

(2) The development and application of the first protein biosensor for any amaryllidaceae alkaloid.

Discussion

The RamR transcription factor was evolved to respond to low micromolar levels of the pathway branchpoint 4NB, and now represents the first protein biosensor for any AA. (p. 13)

(3) The first crystal structure of an amaryllidaceae alkaloid methyltransferase.

In this revision, we determined the first crystal structure of an amaryllidaceae alkaloid methyltransferase, which provides a key mechanistic understanding of where and how beneficial mutations can be introduced into this class of enzymes. The crystal structure confirms the hypothesis that the G40E residue forms a stabilizing salt bridge with the native K118 residue. In addition, docking norbelladine to the newly solved structure reveals that the native K158 and S52 residues likely form hydrogen bonds with this substrate, while the A53M residue sterically positions the 4-hydroxy position of norbelladine for methylation. Together, these results constitute a new **Figure 6**, which is summarized in the **Results** section entitled “Crystal structure of an improved norbelladine methyltransferase” and accompanying **Discussion**.

Figure 6. Crystal structure of an engineered Norbelladine 4-O-methyltransferase (a) Global structure of Nb4OMT^{E36P/G40E/A53M} solved in 2.4Å resolution. One dimer is colored blue while the other dimer is transparent. (b) Comparison of N-termini of Nb4OMT crystal structure (orange) and wild-type Nb4OMT AlphaFold2 structure (blue). (c) Local context of the A53M mutant residue. (d) Local context of the E36P and G40E mutant residues. Black arrow indicates the shift of glutamate from

position 36 to position 40. (e) Active site context of Nb4OMT^{E36P/G40E/A53M} in complex with SAH and docked with norbelladine. For a, c, d, and e, the color coding is as follows - calcium ions: green, s-adenosylhomocysteine: purple, mutant residues: orange, non-mutant residues: gray, docked norbelladine: seafoam green, interactions: yellow dashed lines. (p. 24)

Results: “Crystal structure of an improved norbelladine methyltransferase”

To better understand the mechanism underlying the three beneficial substitutions in the Nb4OMT^{E36P/G40E/A53M} variant, we determined the structure of the Nb4OMT^{E36P/G40E/A53M} variant in complex with S-adenosyl-L-homocysteine (SAH) at 2.4 Å resolution. The Nb4OMT variant exists as a homodimer in the crystalline form (**Figure 6a**), consistent with its size exclusion chromatogram (**Supplementary Figure 13**). The overall fold of the protein was almost identical to the predicted AlphaFold2 structure, except for the N-terminal region (**Figure 6b**). AlphaFold2 predicts that Lys13 forms tight salt bridge interaction with Asp155, Asp181, and Asn182 in the enzyme active site, while the experimental structure showed that Asp155, Asp181, and Asn182 instead coordinate a Ca²⁺ ion and Lys13 forms hydrogen bonds with the backbone of Tyr186 and the sidechain of Tyr194 (**Figure 6b**).

The experimental structure of Nb4OMT^{E36P/G40E/A53M} provides a basis for the improved thermostability of the enzyme (an increase in T_m from 52.8°C to 58.4°C). The A53M substitution inserts a larger hydrophobic methionine inside the hydrophobic pocket formed by Trp50, Tyr81, and Tyr108 (**Figure 6c**), stabilizing the active site of Nb4OMT. The E36P-G40E double mutant shifts a glutamate from position 36 to position 40 and thereby preserves the salt bridge interaction with Lys118 while proline capping the alpha helix (**Figure 6d**).

To better determine how the A53M substitution affects the substrate recognition of Nb4OMT, GNINA 1.0 was used to dock norbelladine into the crystal structure of Nb4OMT^{E36P/G40E/A53M} with SAH and Ca²⁺ already in the active site (based on F_o-F_c electron densities; **Supplementary Figure 14**). In the docked structure, the Ca²⁺ ion positions the catechol moiety of the substrate adjacent to the SAH binding site (**Figure 6e**). A similar substrate recruitment by divalent metal ions is found in other, homologous methyltransferases^{36,37}. A sulfur-π interaction between the catechol group of norbelladine and Methionine 53 may also restrict the rotation of the catechol group, thereby reducing the cross-methylation of the 3' position and improving specificity. (p. 12-13)

Discussion:

The solved crystal structure of the engineered Nb4OMT enzyme provides insights into the stabilization afforded by the E36P/G40E substitutions, and the increased activity and regiospecificity afforded by the A53M substitution. Interestingly, the active site of the solved structure differs from AlphaFold2 models, likely due to lack of the metal ions, ligands, and cofactors. While the recently described AlphaFill tool could potentially address this issue for some models, it did not incorporate the norbelladine substrate or provide a more accurate model of Nb4OMT⁴³ (**Supplementary Figure 15**). Together with the 4NB2 sensor and MutComputeX model, the Nb4OMT variant structure should accelerate progress towards further engineering of the AA pathway. (p. 14)

We have also now added an accompanying **Methods** section for protein purification and crystallography assays, in addition to one **Supplementary Table** and two **Supplementary Figures** describing crystallography metrics, protein chromatography profiles, and the omit FoFc map of the solved crystal structure, respectively.

Major Point #2: Explore the mechanism behind beneficial enzyme mutations

In our revised manuscript we include new data that thoroughly explains the mechanism underlying the key beneficial mutations. We include steady state kinetic data defining the k_{cat}/K_m , k_{cat} , and K_m for the wild-type enzyme, the single A53M substitution, and the triple E36P/G40E/A53M substitution (new **Table 1**). The A53M variant increased k_{cat}/K_m by a factor of 4.7, due to a ~2-fold higher k_{cat} and ~2.5-fold lower K_m for norbelladine compared to the wild-type enzyme. In addition, we characterize the thermal stability of the enzyme variants and show that the single and triple substitutions have between a 1.7 to 5°C increase in T_m . Additions to the “Characterization of improved norbelladine methyltransferase variants” section describe this new data.

Enzyme	k_{cat} (min ⁻¹)	K_m (μM)	k_{cat}/K_m (μM ⁻¹ min ⁻¹)	T_m (°C)
Wild-type Nb4OMT	78 (68 - 92)	122 (82 – 178)	0.64 (0.50 – 0.87)	52.8 (52.8 - 52.8)
A53M mutant	153 (133 – 178)	50 (30 – 82)	3.0 (2.1 – 4.8)	54.5 (54.3 -54.6)
E36P/G40E/A53M mutant	123 (107 - 144)	47 (29 – 73)	2.6 (1.8 – 3.9)	58.4 (58.4 - 58.4)

Table 1. Kinetic and thermal parameters of the wild-type and mutant Nb4OMTs. Lower and upper bounds for the 95% confidence interval from confidence contour analysis for each parameter are given in parentheses. (p. 23)

Results: “Characterization of improved norbelladine methyltransferase variants”

To further understand the mechanism behind beneficial mutations, we characterized the steady state kinetic and thermal properties of NbOMT bearing the A53M substitution alone or in combination with the E36P and G40E substitutions. The A53M substitution increased k_{cat}/K_m by a factor of 4.7, due to a 2-fold increase in k_{cat} and a 2.4 fold reduction in K_m , and increased the T_m by 1.7°C relative to the wild-type enzyme (**Table 1; Supplementary Figure 12**). The Nb4OMT^{E36P/G40E/A53M} triple substitutions appeared to have k_{cat} and K_m values similar to the Nb4OMT^{A53M} single mutant, but a 5.6°C increase in T_m relative to the wild-type Nb4OMT enzyme. These *in vitro* characterization data agree with the *in vivo* data collected with the 4NB-responsive biosensor (**Figure 5a**). (p. 12)

We have also added an accompanying **Methods** section for kinetic and thermal characterization assays.

Major Point #3: Tone down claims regarding generalisability

We would like to thank the reviewers for pointing out this concern. We have since made the following text edits to the **Introduction** and **Discussion** to tone down claims regarding the generalisability of our approach.

Introduction:

Here, we uniquely synergize the development of custom biosensors with machine learning-guided protein design as a paradigm for rapidly prototyping and improving new pathways. to improve microbial fermentation of the branchpoint AA 4-O'Methylnorbelladine (4NB). (p. 4)

Discussion:

While the development of strains for microbial AA biosynthesis is a significant endeavor, we believe that the new 4NB biosensor, ML framework, and enzyme structure presented in this work will significantly accelerate progress towards this goal. ~~approach presented in this work, synergizing custom biosensor enabled screens with self-supervised machine learning guided protein design, will fundamentally accelerate the pace of strain and enzyme engineering as a whole.~~ (p. 15)

Reviewer #1:

Remarks to the authors:

"In the presented study the authors show-case a follow-up study based on their previous work on RamR as an evolvable biosensor-scaffold useful for high-throughput screening of relevance to biocatalysis optimization. Specifically, in this study they demonstrate the RamR versatility by finding RamR-variants able to detect Amaryllidaceae alkaloids in particular towards the common precursor 4NB. The authors used their previously developed toggle-selection regime SELIS to develop a biosensor for 4NB with a detection limit of 2.5 μ M and 80-fold selectivity for 4NB over its precursor NB. With a biosensor at hand the authors aimed to use it for screening of Nb4OMT enzyme variants designed based on machine learning to accelerate the discovery of enzymes and pathways with improved stability and activity.

To do so they rebuilt the data engineering pipelines from MutComputeX to accept cofactors and ligands for Nb4OMT, demonstrating superior feature extraction capabilities compared to previous model. Ultimately the authors demonstrate the identification of Nb4OMT variant enzymes with up to 60% improvement in product formation compared to wild-type Nb4OMT, while the level of substrate norbelladine decreased 17-fold, while off-product decreased by approx. 3-fold.

The study convincingly makes use of a very powerful approach combining directed evolution and machine learning for metabolic engineering/biotech applications, and with demonstrated success both in terms of biosensor engineering and application. Furthermore, being a common precursor to a multitude of AAs, the 4NB biosensor should be useful for further pathway discovery and 4NB optimization efforts. This also goes for the Nb4OMT variants.

However, the general approach to directed evolution of biosensors, and the ML-/AlphaFold-guided enzyme variant library design is not novel. Neither is the application of the biosensor used for functionalizing upstream 4NB enzymes, alike norcraugsodine reductase, and norbelladine synthase, never previously expressed in microbes."

While directed evolution has been performed previously, the melding of directed evolution and computational prediction approaches for pathway optimization is a novel and important direction, bringing out the features of both. In particular, while AlphaFold2 structural models have been used to guide rational enzyme design by others, these structures have never before been used as inputs to machine learning models for enzyme engineering; all previous examples have used solved crystal structures as inputs for machine learning approaches. The use of directed evolution for biosensor

development, and the use of computational predictions to rapidly exploit the capabilities of the biosensor is overall an important milestone in how to combine technologies to advance pathway engineering, resulting in one of the first improvements in the early steps of AA biosynthesis. To clarify the novelty of our combined approaches, we have made text additions to the **Results** and **Discussion** (please refer to **Major Point #1**)

“Likewise, no generally applicable mechanistic understanding of the engineered MTase is provided. These points all limit the scope of the work presented beyond the impactful previous studies from this team. According to this reviewer, this precludes publishing the study in this journal. This said the optimization of MutComputeX is very interesting, and so are the novel RamR and enzyme variant, and the study should be suitable for a more specialised journal focusing on method development for metabolic/enzyme engineering or computational biology.”

We have since collected structural and kinetic data on the improved Nb4OMT triple mutant. The new X-ray crystal structure, described in the new **Figure 6**, yield a mechanistic explanation for the key role of the A53M mutation in improving catalytic activity by 4-fold, and an in-depth description of this explanation is found in the new **Results** section entitled “Crystal structure of an improved norbelladine methyltransferase.” Please again refer to **Major Point #1**, which fully describes the new Nb4OMT variant’s structure and kinetics.

Major comments/concerns:

“In the first Result paragraph it states that norbelladine was docked on to RamR, yet both legends in Fig 1c and Suppl. Fig. 1 states 4NB. In the second Result paragraph it (correctly?) states 4NB-based docking to inform library design. Please explain or correct.”

We thank this reviewer for bringing this discrepancy to our attention. The **Results** section was edited to correct this mistake.

Results: “Identifying a biosensor for the branchpoint amaryllidaceae alkaloid 4-O’Methyl-norbelladine To better understand this promiscuous binding activity, 4NB was docked within the ligand binding pocket of RamR using GNINA ... (p. 5)

“In relation to Fig. 2 it is unclear to this reviewer what difference there is between panel b) and f) when it comes to RFU/OD. In both panels bionseor 4NB2.1 is used yet panel b) has a signal of 0.5×10^4 , while panel f) shows 0.6×10^5 RFU/OD. Please explain the shift in signal. In relation to these two plots it would also be needed to further elaborate on the operational range of the 4NB2.1 biosensor. In panel b) it seems to saturate at approx 100 uM 4NB, while in panel f) it is evident that a (significant?) difference can be observed between 100 and 250 uM.”

We greatly appreciate the keen observation made by this reviewer. The difference in relative fluorescence in **Figure 2b** compared to **2f** is due to a difference in the amount of time the cells were incubated with the ligand; for **Figure 2b**, cells were incubated with 4NB for 4 hours, the standard assay time we use for biosensor characterization. In **Figure 2f**, cells were incubated with 4NB for 18 hours, since these are the same conditions employed when the 4NB2 sensor is used to measure enzyme activity (as in **Figure 3**; a longer incubation time allows for the enzyme-catalyzed formation of 4NB). This difference also explains the slight change in operational range between **Figure 2b** and **Figure 2f**, as

pointed out by this reviewer. To clarify this discrepancy, the following text edits were made to the manuscript:

Legend of Figure 2

“... All data was performed in biological triplicate. Cells were cultured for 4 hours with the ligand in b & c, and for 18 hours with the ligand in f & g. Error bars represent the S.D. +/- the mean. ...” (p. 19)

Methods section “Biosensor response assay”

Cultures were grown for an additional 4 h at 37 °C and 250 r.p.m. and subsequently centrifuged (3,500g, 4 °C, 10 min), except in the case of the HPLC comparison assays (Figure 2f,g), where cultures were grown for an additional 18 hours at 37 °C and 250 r.p.m. in order to compare sensor response in fermentation-relevant conditions. (p. 31)

“On the note “...it failed to enhance Nb4OMT activity”, from the epPCR variant library screen, could the authors provide speculations or data indicating why the phenotypes from colonies on plates to population/single cell fluorescence from liquid cultures did not correlate? Plate screens are very powerful, and dirt cheap. Readers would benefit from the authors’ considerations here.”

We apologize for the confusion. We did not directly compare the fluorescence of colonies on plates to the fluorescence of cell populations in liquid culture. We picked the brightest colonies on plates into liquid culture, but at this stage we were not able to readily quantitatively compare the fluorescence of these colonies to cells expressing the wild-type enzyme. In the subsequent population measurements with liquid cultures we did compare fluorescence produced by the mutant enzymes to the wild-type enzyme and found no significant difference. In fact, many of our top variants recovered from this screen yielded enzymes with the same exact genotype as the wild-type enzyme. This result highlights the need for computational approaches to more rapidly advance enzyme engineering.

To improve clarity for the reader, we have made the following edit in the results section:

The library of enzyme variants was transformed into cells containing the pSens4NB2.1 plasmid, plated on solid media containing norbelladine, and highly fluorescent colonies were isolated and then individually phenotyped in a secondary, quantitative liquid-based fluorescence screen where they were compared to the wild-type enzyme. Unfortunately, this approach was not able to identify variants that outperformed the wild-type enzyme in the liquid-based screen. (p. 8)

“Since epPCR library size of Nb4OMT is not estimated/indicated and that no variants with enhanced activity was identified from the plate assay, it is unclear to this reviewer why the authors move on to adopt machine learning based on structure-guided engineering.”

We agree that the rationale for adopting a machine learning-based approach could be clarified to the reader. While the plate-based approach may have eventually yielded improved enzyme variants (as we have shown previously, d'Oelsnitz S, Kim W, et al. 2022), it requires a very idiosyncratic development of libraries and selection pressures, and likely has to be reoptimized for each new enzyme. The power of machine learning is that when it is pre-trained on available enzymes (as MutComputeX is), it ‘pre-learns’ what is likely a good or bad substitution for a given position, thus greatly reducing the number of possibilities that need to be tested, and (with an appropriate biosensor) allowing a rapid screen that may proceed more efficiently than an unoptimized selection. In fact, the machine learning-based approach could find mutations that were unlikely to occur in the error-prone library. For example, the beneficial

A53M mutation identified by our AI model required two mutations (GCG → ATG), which would be extremely rare in an epPCR library. Accordingly, we have made the following edits to the results section to clarify this rationale, and provide detail on the epPCR size:

To improve Nb4OMT activity in a microbial host we initially carried out directed evolution starting from randomly mutagenized libraries, via error-prone PCR, which generated an average of three mutations per gene. (p. 8)

To pursue a complementary approach to enzyme engineering, we sought to use machine learning to guide enzyme design, an approach that could identify variants unlikely to occur via random mutagenesis. (p. 8)

“The plate-screen is claimed to yield different fluorescence intensities (even though this data is not presented), and if this was the case why would colonies from this assay not form relevant training data for ML?”

The machine learning models used in this study are self-supervised models, which is a type of unsupervised learning. Thus, the architecture of the model is not structured in a way that could use the colony assay data for the training process.

“It is an over-statement to say that a structure-based CNN-based approach was pursued when no Nb4OMT structure is available to support the training data. Authors should rather state that CNN was considered but aborted due to lack of structural data.”

We thank this Reviewer for bringing up a potential source of confusion, as the reviewer essentially makes the case regarding the novelty of our results. It is true that no Nb4OMT structure was solved when pursuing enzyme engineering; however, we were able to create an AlphaFold2 structural model of the Nb4OMT protein docked with SAH and norbelladine, and used this as the input to our model. Therefore, we were able to leverage recent advancements in machine learning structure prediction (AlphaFold2) and ligand docking (GNINA) to “fill in the gap” when no crystal structure data was available; a point we now emphasize in terms of the novelty of our results (**Major Point #1**). We have also made the following text edits to the **Results** section.

To produce MutComputeX-guided designs, we generated a Nb4OMT enzyme structure file to serve as an input to the model. Unfortunately, Although the structure of the Nb4OMT enzyme had not been solved; preventing the generation of structure-based CNN predictions for substitutions we were able to create a *de novo* structural model for Nb4OMT was generated using AlphaFold2²⁶, which was then docked with both the S-adenosyl-homocysteine (SAH) cofactor and norbelladine using GNINA1.0²⁵. (p. 10)

Minor comments/concerns:

“L3 in Intro. Please insert what AA stands for - Amaryllidoideae alkaloids?”

The following text edit was made in the Introduction to clarify what AA refers to:

Among the approximate ~600 reported **Amaryllidoideae alkaloids (AAs)**, those derived from the lycorine, haemanthamine, and narciclasine scaffolds have been used as lead molecules in anticancer research . One of the most notable **Amaryllidoideae alkaloids (AAs)** is galantamine, a selective ... **(p. 3)**

“Legend for Suppl. Fig. 2 needs updating on numbering of subpanels. Also, no indication of replicates is provided in the figure legend. Please update this, incl indication of error bars as well as statistical testing if applied.”

We greatly appreciate the observation made by this Reviewer. We have since updated the legend for **Supplementary Figure 2**.

Supplementary Figure 2: The sensitivity and selectivity of RamR mutants evolved for 4-OMe-norbelladine. (a) Dose response measurements and genotypes of generation one RamR sensors. (b) Selectivity of generation one RamR sensors. (c) **Dose response measurements and genotypes of generation two RamR sensors.** (d) **Selectivity of generation two RamR sensors.** Experiments were conducted in biological triplicate. Error bars represent the S.D. +/- the mean. **(p. 3)**

Reviewer #2:

“This manuscript is focused on developing better ways of biocatalytic production of 4-O’Methylnorbelladine (4NB) from norbelladine (Nb) via directed evolution and machine learning-based protein engineering. The first part covers two rounds of directed evolution of the RamR repressor for detecting 4NB, based on three hotspots identified by docking 4NB and the SELIS method published earlier. These two rounds led to a significantly more sensitive (4NB) and specific (4NB vs Nb) variant, which was then coupled with Nb 4-O-methyltransferase (Nb4OMT) in E. coli to create an in vivo assay for activity measurement, used for directed evolution of Nb4OMT in the second part.

For evolving Nb4OMT, the authors first applied random mutagenesis, according to the authors, with little success. Then they modelled the Nb4OMT structure with AlphaFold2, docked Nb and a cofactor, and applied a modification of MutCompute (retrained on protein-ligand complexes) to suggest substitutions. After experimentally testing 22 mutants manually selected based on the MutCompute output, the authors ended up with several improved designs.

In general, the study covers an impressive workflow consisting of biosensor development, self-supervised learning, and protein engineering. The results are novel in each part, and the study will definitely benefit biochemists and protein engineers. However, while the work is exciting, the reporting sometimes lacks clarity, and some of the claims seem too strong for the evidence provided in the text. I outline my reservations below.”

We thank this Reviewer for their interest in our study, and we appreciate their comments to help improve the clarity and quality of this work.

Major comments/concerns:

“Directed evolution of the 4NB sensor:

After the first round of the directed evolution targeting sensitivity for 4NB, the authors chose 4NB1.2 as a starting point for the second round, which targeted counter-selection against Nb. But based on the data shown in Figure S2, 4NB1.1 and 4NB1.3 had higher sensitivity and seemed to be better candidates for the second round. Moreover, 4NB1.2 already showed close-to-zero sensitivity for Nb, so evolving it for the counter-selection against Nb seems counterintuitive. Could the authors please comment on why they preferred 4NB1.2?"

This observation raises an important point around how to best decide on the next variant for additional rounds of directed evolution. In reality, there may not always be an obvious protein variant to pursue. This reviewer is correct in that 4NB1.1 and 4NB1.3 were more sensitive than 4NB1.2. However, these two variants were also less selective for 4NB compared to the 4NB1.2 variant. Since selectivity is important for accurately monitoring pathway activity and was absolutely required to pursue subsequent engineering of the Nb4OMT enzyme, we chose to use 4NB1.2 as the template for the second round of evolution. To clarify this rationale, we have made the following edits to the results section of the manuscript:

In fact, one variant bearing two amino acid substitutions (4NB1.2, K63T and L66M) displayed a 20-fold selectivity for 4NB over norbelladine (Supplementary Figure 2a, b). Although two other RamR variants had greater sensitivity for 4NB, the higher selectivity of the 4NB2.1 variant rendered it more suitable for accurately monitoring pathway activity. Using 4NB1.2 as a starting point, additional libraries were generated that encompassed the other, previously randomized positions. (p. 6)

"The authors highlight the superiority of the 4NB sensor over HPLC based on the lower dynamic ranges for the concentration of 4NB (2.5 μ M -- 250 μ M vs. 25 μ M -- 1000 μ M) and refer to Figures S2c, d and Figure 2f as evidence. However, those figures are not convincing in this respect as they do not seem to cover the low concentration values in detail."

Although **Figure 2f** does not cover lower concentration values (2.5 - 25 μ M), this data was in fact collected but was left out to improve clarity since the HPLC method cannot distinguish between these concentrations. However, the biosensor can distinguish these concentrations, as shown in **Figure 2b** and **Figure S2c**. To clarify this to the reader, we have made the following text edits and now include a new **Supplementary Table** containing HPLC measurements and error values for all tested concentrations.

The concentration range of 4NB can be discerned between 2.5 μ M and 250 μ M, while the equivalent range for the HPLC method is between 25 μ M and 1000 μ M (Figure 2f) (**Supplementary Table 1**). (p. 7)

"What was the 4NB concentration for the measurements reported in Figures S2 b,d?"

The legend of **Supplementary Figure 2** has been updated to include the 4NB concentrations used for measurements in **Figure S2b** and **S2d**. We thank this reviewer for bringing this to our attention.

Supplementary Figure 2: The sensitivity and selectivity of RamR mutants evolved for 4-OMe-norbelladine. (a) Dose response measurements and genotypes of generation one RamR sensors. (b) Selectivity of generation one RamR sensors. (c) Dose response measurements and genotypes of generation two RamR sensors. (d) Selectivity of generation two RamR sensors. For measurements performed in (b, d), cells were cultured with 100 μ M of 4-OMe-norbelladine. Experiments were conducted in biological triplicate. Error bars represent the S.D. +/- the mean. (p. 3)

“Directed evolution of Nb4OMT:

The authors mention that the new self-supervised 3DResNet achieved improved wildtype prediction accuracy 80% compared to 69% for the previously published version. However, after ensemble-based fine-tuning using the data from FireProtDB, the protein-ligand interface wildtype accuracy dropped to 66%. Why not use the original model then?”

Although the Reviewer is correct in pointing out that the wild-type accuracy of the ensemble 3DResNet is lower than the non-ensembled net, we chose to use the ensemble net largely because it correlated better with the experimentally collected Δ TM point mutation data from FireProtDB. In other words, it was possible that the non-ensembled net was overtrained to predict wild-type residues, and we wished to predict improved variants. Bolstering this interpretation, we have observed that improved wild-type prediction accuracy does not necessarily translate to improved protein engineering ability. To clarify our rationales in the manuscript, we have made the following edit to the Results section:

The ensembled 3DresNet model (MutComputeX) had an overall wildtype accuracy of 67.3% and protein-ligand interface wildtype accuracy of 66%. While the wildtype accuracy of MutComputeX is lower than what is capable by the 3DResNet framework, we chose to use this model since it correlated best with experimentally collected data from FireProtDB³⁴. (p. 10)

“It is striking that only one mutation showed a consistent increase in the signal at low temperatures because in other studies using MutCompute, the success rate was much higher (8-9 out of 10). Can it be that the AlphaFold2-generated structure is unreliable?”

We have observed the success rate of MutCompute unsurprisingly differs between enzymes. For example, in previous efforts with *Bst* DNA polymerase five of the top ten designs displayed activities equal to or greater than the parent enzyme (Paik, I. et al., *Biochemistry*, 2023), while in work on a plastic-degrading enzyme, PETase, eight of the top 18 mutations improved enzyme activity (Lu, H. et al., *Nature*, 2022). In the current study, we’ve found that five of the 22 MutComputeX-guided single point mutations improved product formation (as indicated by fluorescence) (**Supplementary Figure 8**). For all of these predictions, the hit rate was such that we could readily assess individual synthetic variants much more quickly than if a library selection or screening method was attempted. Overall, our experience is that the variation in the success rate of AI-guided designs is more closely associated with the protein target. This is not surprising, as natural phylogenies also readily suggest that different proteins have different evolvabilities.

As to whether or not the AlphaFold2 structure is unreliable, since solving the crystal structure of the Nb4OMT triple mutant, we have found that the AlphaFold2 structure is consistent with the solved structure with the exception of the first nine amino acids in the N-terminus. It is possible that this minor discrepancy might have contributed to the lower success rate of our AI-guided designs, but we were hesitant to speculate on this in the manuscript itself.

“As I understand, the mutant selection for experimental validation involved significant manual curation of MutComputeX output. This should be mentioned explicitly in the results and discussion, and the selection deserves more details than just a short sentence at the end of Mutational Designs. ...”

and

““MutComputeX ... predicted functional variants of the Nb4OMT enzyme with 60% improvement in product titer, 17-fold reduced remnant substrate, and 3-fold lower off-product formation” should be rephrased as “MutComputeX... allowed constructing several variants with 60% improvement ...”.”

This Reviewer correctly points out that the manual curation process should be clarified. While we have included a sentence in the Results section briefly describing this process, we have since provided more detail about the curation process in a new **Supplementary Discussion**, which we now reference in the **Results** and quote below.

Results: “Developing a machine learning pipeline for structure-based enzyme engineering “

Based on these predictions, we manually curated predicted substitutions, prioritizing those that were near the active site and that were likely to form known stabilizing motifs, such as salt bridges. Our rationale for choosing the selected mutations is provided in **Supplementary Discussion 1. (p. 10)**

Supplementary Discussion:

A53:

ALA: 0.01 MET: 0.37 ARG: 0.38

MutComputeX prefers MET in the presence of ligand/cofactor, and it prefers ARG with no ligand/cofactor. ARG and MET may form either a Cation-pi interaction or a Sulfur-pi interaction with the catechol ring of norbelladine, respectively

Rank: MET > ARG

S159:

SER: 0.05 GLU: 0.44

MutComputeX strongly predicts GLU at position 159.

Rank: GLU (**p. 29-30**)

In addition, we have included text in the **Discussion** to reiterate the fact that a manual curation process was indeed a component of protein variant design.

MutComputeX was trained on ~2.3M microenvironments sampled from over 23,000 protein structures, and facilitated the manual curation of variants with a 60% improvement in product titer, 3-fold lower off-product formation ... (**p. 14**)

Finally, we have also referenced the new **Supplementary Discussion 1** in the “Mutational Designs” section in the Methods:

Mutations were designed with two goals: stabilizing the protein away from the active site and investigating point mutations where predictions differed between the docked and apo protein structures. With these objectives, we sorted residues based on the log odds between the predicted and wild type amino acids. For the stability objective, predictions that recapitulate known chemical

phenomena such as salt bridges, hydrogen bonding, proline capping were prioritized. An in-depth discussion of the mutation curation process is described in **Supplementary Discussion 1. (p. 37)**

... Also, the statement “We believe that the approach presented in this work, synergizing custom biosensor-enabled screens with self-supervised machine learning-guided protein design, will fundamentally accelerate the pace of strain and enzyme engineering as a whole” is an exaggeration as the study does not demonstrate the generalisability of the protocols to other systems. The manual curation step implies that the approach is quite case-study-specific.”

We agree with this Reviewer that the wording in the **Discussion** should be toned down (please refer to **Major Point #3**).

“I could not find any information on the numbering of OMT variants in figures, e.g., see Figure 4b. Could the authors please provide a table with the exact substitutions introduced to each variant?”

A new **Supplementary Table 3** was created that describes the exact substitutions made in each variant shown in **Figure 5b** (previously **Figure 4b**). We thank the Reviewer for pointing out this source of confusion.

Mutant Alias	Genotype
17-203	H17K, V203E
17-36	H17K, E36P, G40E
17-159	H17K, S159E
36-203	E36P, G40E, V203E
17-53	H17K, A53M
17-53-203	H17K, A53M, V203E
17-53-159	H17K, A53M, S159E
53-159	A53M, S159E
53-203	A53M, V203E
36-53	E36P, G40E, A53M
36-53-203	E36P, G40E, A53M, V203E
36-53-159	E36P, G40E, A53M, S159E

Supplementary Table 3: Genotypes and aliases of combinatorial Nb4OMT mutants

In addition, the legend of **Figure 5** was edited to reference the new **Supplementary Table**.

Figure 5. In vivo characterization of ML-designed Nb4OMT variants

(a) Fluorescent signal produced from E. coli cells containing the 4-OMe-norbelladine reporter plasmid (pSens-4NB2) and expressing either an empty plasmid (TAA), the wild-type Nb4OMT enzyme (WT), or Nb4OMT mutants, when cultured with 100 μ M of norbelladine at 37°C. The blue horizontal line denotes the fluorescent signal produced from culturing the wild-type Nb4OMT enzyme. Error bars represent the S.D. +/- the mean. **Genotypes of all variants can be found in Supplementary Table 3. (p. 22)**

“Training of MutComputeX:

How many atom channels were introduced to the original architecture because of the ligands?”

To enable generalization to ligands, nucleotides, and cofactors, we added a channel for phosphorus atoms and a halogen channel that has all F, Cl, Br, I atoms. While these additional channels do not impact predictions for Nb4OMT in particular, they are part of the new MutComputeX package that allows generalization to structures beyond proteins. To clarify this, we have made the following changes to the **Results** section and the **Methods** section:

Results

To develop MutComputeX, we first rebuilt the data engineering pipelines to enable training on heterogenous microenvironments (**Figure 4b**). New atomic channels for phosphorus and grouped halogens were added to the input representation (see **Methods**). (p. 9)

Methods

Each atom is placed into a corresponding element channel except halogen atoms (which are placed into a multi-atom channel that consist of F, Cl, Br, I), resulting in the following atomic channels: C, H, O, N, S, P, Halogens. Each atom’s partial charge and SASA value are placed into the partial charge and SASA channels, respectively. For all 9 channels, atom values are gaussian blurred according to their Van-der-Waals radii. The P and Halogen channels were added to the original MutCompute framework in order to generalize to ligands and nucleic acids. (p. 35)

Additionally, the new MutComputeX Figure clearly describes all channels used (please refer to **Major Point #1**).

“This provided us with X protein sequences from Y PDB entries.” What are X and Y?”

We thank the Reviewer for pointing out this error. We have updated the “Dataset Generation” section in the **Methods** as follows:

This provided us with 22584 protein sequences from 21613 PDB entries. (p. 36)

“How was the prioritization of residues within 5Å of non-protein entities implemented?”

Prioritization was performed using Cartesian space calculated by the Contact Search functionality of the GEMMI library. We have further described this process in the “Dataset Generation” section of the **Methods** section:

To generate microenvironment datasets, for each protein chain we prioritized residues that were within 5Å of a non-protein entity via the GEMMI library ContactSearch functionality, and then randomly backfilled until 200 residues or half of the protein sequence was sampled. (p. 36)

“Did the authors consider calculating the correlations with FireProtDB data for the scores returned by the original MutCompute method?”

We would like to thank the reviewer for pointing out this potential source of confusion. Yes, these values were included in **Supplementary Table 6** indicated in the row labeled “3DCNN ensemble”. We have clarified this in the legend of **Supplementary Table 6**:

Supplementary Table 6: Wildtype accuracy and Pearson and Spearman correlation metrics with ΔT_m and $\Delta\Delta G$ experimental values for point mutations in FireProtDB2 (as of February 2022). Correlations were calculated with the log odds predicted by the deep learning model for the mutated and wildtype amino acids: $\log(\text{mutAA_probability}/\text{wtAA_probability})$. The original MutCompute model is labeled as “3DCNN ensemble”, while the MutComputeX model is labeled as “3DResNet ensemble”. (p. 24)

Minor comments/concerns

“In many graphs, the authors mention that error bars represent the S.E.M +/- the mean, but they look more like standard deviations to me. In fact, I would report the latter for triplicates.”

We would like to greatly thank this reviewer for making this keen observation. Upon reviewing the methods used for calculating error, we can confirm that the error bars do in fact represent standard deviations. We have since edited the legends of **Figure 1**, **Figure 2**, **Figure 5**, **Supplementary Figure 2**, **Supplementary Figure 4**, **Supplementary Figure 5**, **Supplementary Figure 8**, and **Supplementary Figure 9**.

“In the first paragraph of the results, the authors mentioned the docking of Nb at RamR. Did they mean 4NB? (based on Figure 1C and the next paragraph).”

We thank this reviewer, in addition to **Reviewer #1**, for bringing this discrepancy to our attention. The **Results** section was edited to correct this mistake. Please refer to our response to **Reviewer #1**, major comment #1 for the text edits we have made.

“The authors mention that random error-prone PCR failed to enhance Nb4OMT activity. It will be valuable to see what activity (fluorescence) distribution the authors observed in this step to have a better idea of the baseline.”

These experiments were carried out without attention to quantitation, and once we found that we were unable to recover any improved enzyme variants, we moved on to the MutComputeX-guided designs. We do have sequencing data for the five most promising Nb4OMT variants that came from the error-prone library screening, which revealed that 3 of the 5 variants had the exact same amino acid sequence as the wild-type Nb4OMT enzyme while the other two (T75A and M143T) produced a signal equivalent to that produced by these wild-type variants. These results underscore that the variants in general were likely not better than wild-type.

“Figure S5 is somewhat confusing. The text under the x-axis (Fed substrate μM) seems to refer to the legend rather than the axis. Could the authors comment on why the concentrations of the substrate and product do not sum up to the input concentrations?”

We thank this reviewer for bringing this error to our attention. Upon further inspection of the raw data, we have also found that the decimal point was off for the reported concentrations of substrate and product. We have since fixed the values and legends in **Supplementary Figure 5**, and have made corrections to the accompanying legend. We have also scrutinized all values to ensure that this mistake was not made elsewhere.

We believe that the concentrations of the substrate and product do not sum up to the input concentration due to the degradation of the substrate. The norbelladine substrate contains a catechol moiety, which is prone to oxidation.

Supplementary Figure 5: Substrate and product of the *in vivo* Nb4OMT reaction measured with HPLC. *E. coli* cells expressing the wild-type Nb4OMT enzyme were cultured with varying amounts of norbelladine for 18 hours and the concentrations of norbelladine and 4-OMe-norbelladine were subsequently measured. Colors denote the concentration of norbelladine supplemented in the culture media during the reaction. Some amount of norbelladine is assumed to be lost during the reaction due to oxidation and degradation. Measurements were performed in biological triplicate and error bars represent the S.D. +/- the mean. (p. 6)

“Many abbreviations and names of the tools are spelled differently throughout the manuscript and should be unified (AlphaFold2 – Alpha fold, Nb4OMT – OMT - 4OMT, etc).”

Abbreviations throughout the paper were standardized as “AlphaFold2” and “Nb4OMT”, including edits to the title of **Figure 3**, **Methods**, and **Results**. We thank this Reviewer for their incredible attention to detail, which has greatly improved the quality of the manuscript.

Reviewer #3:

“Amaryllidaceae alkaloids are a special group of alkaloids presented in the plants of the family Amaryllidaceae. Some of them such as galanthamine are well known for its therapeutic application for the patients with Alzheimer’s disease. Generally speaking, it is very interesting and challenging to reconstitute a whole biosynthetic pathway in *E. coli* to produce the specific plant natural product of interest. In the present study, the authors claimed a synthetic microbial sensing and biosynthesis of Amaryllidaceae alkaloids. Directed evolution of RamR, a starting point for identifying biosensors for a variety of benzylisoquinoline alkaloids, led to a specific biosensor for 4'-O-methyl-norbelladine. However, directed evolution of the O-methyltransferase Nb4OMT using the error-prone PCR for saturated mutagenesis did not enhance its catalytic activity. Thus the authors developed an AI technology model MutComputeX to generate catalytic activity-enhanced mutant for Nb4OMT. The experimental results showed a 60% improvement of the yield of 4'-O-methyl-norbelladine, 17-fold reduction of remained substrate, and 3-fold lower of the production of 3'-O-methyl-norbelladine.”

We thank this reviewer for their interest in our work.

“All results are very interesting. However, the novelty of the whole study is very limited.”

As we detail previously (**Major Point #1**), we create the first structure-based machine learning model that has been explicitly trained to generalize to non-protein atoms and uniquely pair it with a novel biosensor, which represents the first protein biosensor for any amaryllidaceae alkaloid, to achieve significant improvements in the production of AA intermediates. To further emphasize the novelty and significance of this work, we have added new structural and kinetics data, which provide a mechanistic explanation for the improved activity and regioselectivity of our engineered enzyme variants. In addition to text edits, we have created a new Figure that describes the AI data structure and architecture, and a new main Figure describing the crystal structure of the engineered Nb4OMT enzyme.

“4'-O-Methyl-norbelladine has been thought to be the branch point of the Amaryllidaceae alkaloids biosynthesis. It is very important to produce 4'-O-methyl-norbelladine with high yield in E. coli. However, it is not difficult to chemically synthesize 4'-O-methyl-norbelladine with high yield. In the biosynthesis of Amaryllidaceae alkaloids, the O-methylation is not the rate-limiting reaction step. Additionally, it is not the entire pathway for 4'-O-methyl-norbelladine but only one enzyme, i.e., Nb4OMT, was investigated in the present work.”

We agree with this reviewer that it is not difficult to chemically synthesize 4'-O-methyl-norbelladine. However, the goal of our study was not simply to enable production of one chemical, rather it was to provide a machine learning framework for enzyme engineering and a biosensor-based approach for screening to accelerate metabolic engineering of Amaryllidaceae alkaloids in microbial hosts. Nonetheless, to more accurately reflect our work we have changed the title of this manuscript to the following:

Title

Synthetic microbial sensing and biosynthesis of a branchpoint amaryllidaceae alkaloid

“Directed evolution is a useful method to enhance the catalytic activity of enzyme. RamR had been applied in identification of biosensor for some alkaloids. The directed evolution of RamR led to a specific biosensor for 4'-O-methyl-norbelladine. However, it failed to enhance the catalytic activity of Nb4OMT. In the present study the authors employed error-prone PCR to mutant Nb4OMT. More rational directed evolution methods may give good results.

MutComputeX model was developed to guide the evolution of Nb4OMT. The results are kind of good. Especially, the A53M mutant is the better one. Is there some reasonable explanation for this mutation?”

In our revised manuscript, we describe new kinetic and structural data that better explains the mechanism behind key beneficial mutations. Our kinetic data shows that the A53M mutation increases the k_{cat} ~2- fold and lowers the K_m ~2.5-fold for the norbelladine substrate relative to the wild-type Nb4OMT enzyme. This data is now presented in **Table 1**. Our new structural data, alongside docking analysis, reveals that the 53M residue is positioned between the sulfur group of the SAH cofactor and the 4'-hydroxyl group of norbelladine, stabilizing the active site configuration while also securing norbelladine's catechol group to hinder the cross-methylation of the 3'-hydroxy group and improve regioselectivity. This data is presented in the new **Figure 6** (please refer to **Major Point #1** for images of **Table 1** and **Figure 6**).

Other updates:

To reflect our new kinetic and crystallography results in the **Abstract** and **Introduction**, we have made the following edits:

Abstract

... A structure-based residual neural network (MutComputeX) was subsequently developed and used to generate activity-enriched variants of a plant methyltransferase, which were rapidly screened with the biosensor. Functional enzyme variants were identified that yielded a 60% improvement in product titer, 4-fold higher catalytic activity, and 3-fold lower off-product regioisomer formation. A solved crystal structure elucidates the mechanism behind key beneficial mutations. (p. 2)

Introduction

... We then developed MutComputeX: a structure-based self-supervised residual neural network (3DResNet) trained to generalize at protein:non-protein interfaces, which was used to generate activity-enriched Nb4OMT designs from an ML-generated protein-cofactor-substrate structure. The evolved biosensor was used to rapidly screen a panel of MutComputeX-guided Nb4OMT designs, leading to the identification of one variant that yielded a 60% improvement in product titer, 4-fold higher catalytic activity, and 3-fold lower off-product formation. A newly solved crystal structure of this engineered enzyme helped elucidate the mechanism behind key beneficial mutations and highlighted important discrepancies with the AlphaFold2 model. (p. 4)

We would also like to note that since our paper was submitted, an exciting pre-print was published in bioRxiv from the Sattely group that discovered the final three enzymes of the galantamine biosynthesis pathway. Accordingly, we have made the following changes to the Discussion section to reference these new findings:

With regards to galantamine, the CYP96T1 and CYP96T6 cytochrome proteins have been shown to catalyze the para-ortho coupling of 4NB and thus produce the galantamine precursor N-demethylnarwedine^{51,52}. Finally, while writing this paper, the remaining NtNMT methyltransferase and NtAKR1 ketone reductase enzymes necessary for complete galantamine biosynthesis have recently been discovered, unlocking the exciting opportunity to achieve the biosynthesis of galantamine from simple feedstocks using a microbial host⁵². (p. 15)

REVIEWER COMMENTS

Reviewer #1 (Remarks to the Author):

d'Oelsnitz et al rebuttal

Comments to major points raised by the authors:

Major Point #1:

This reviewer agrees that the authors have substantially improved the understanding of the workflow for MutComputeX and also clarified the extension from models based on MutCompute. This said, and without any means to discredit the excellent work of the authors on the sampling of high-quality data used for training, the interim release of AlphaFold v2.3 since the first submission makes this reviewer doubt that the authors can claim “first-in-class” (i.e. “To create designs that could be generalized to protein-ligand interfaces, we have developed MutComputeX: the first structured-based neural network designed to generalize to microenvironments at protein-ligand”). As a general note, this reviewer finds that such statement should be avoided.

Major Point #2:

Similarly, in relation to the novel biosensor, the statement that the evolved RamR variant represents “...the first protein biosensor for any AA. (p. 13)” should be avoided, and considered to be replaced by evidence-based conclusions, e.g. “...represents to the best of our knowledge a first protein-based biosensor for detection of a amaryllidaceae alkaloid”.

Fig. 4 legend: There is something wrong/unclear with respect to the legend of the new Figure 4c-d. The description of the color codes does not relate to the colors of the spheres in the figure.

Below are my individual responses to the author's rebuttal and revised manuscript.

Reviewer #1:

Remarks to the authors:

“In the presented study the authors show-case a follow-up study based on their previous work on RamR as an evolvable biosensor-scaffold useful for high-throughput screening of relevance to biocatalysis optimization. Specifically, in this study they demonstrate the RamR versatility by finding RamR-variants able to detect Amaryllidaceae alkaloids in particular towards the common precursor 4NB. The authors used their previously developed toggle-selection regime SELIS to develop a biosensor for 4NB with a detection limit of 2.5 μ M and 80-fold selectivity for 4NB over its precursor NB. With a biosensor at hand the authors aimed to use it for screening of Nb4OMT enzyme variants designed based on machine learning to accelerate the discovery of enzymes and pathways with improved stability and activity. To do so they rebuilt the data engineering pipelines from MutComputeX to accept cofactors and ligands for Nb4OMT, demonstrating superior feature extraction capabilities compared to previous model. Ultimately the authors demonstrate the identification of Nb4OMT variant enzymes with up to 60% improvement in product formation compared to wild-type Nb4OMT, while the level of substrate norbelladine decreased 17-fold, while off-product decreased by approx. 3-fold.

The study convincingly makes use of a very powerful approach combining directed evolution and machine

learning for metabolic engineering/biotech applications, and with demonstrated success both in terms of

biosensor engineering and application. Furthermore, being a common precursor to a multitude of AAs, the 4NB biosensor should be useful for further pathway discovery and 4NB optimization efforts. This also goes for the Nb4OMT variants.

However, the general approach to directed evolution of biosensors, and the ML-/AlphaFold-guided enzyme variant library design is not novel. Neither is the application of the biosensor used for functionalizing upstream 4NB enzymes, alike norcraugsodine reductase, and norbelladine synthase, never previously expressed in microbes.”

>> While directed evolution has been performed previously, the melding of directed evolution and computational prediction approaches for pathway optimization is a novel and important direction,

bringing out the features of both. In particular, while AlphaFold2 structural models have been used to guide rational enzyme design by others, these structures have never before been used as inputs to machine learning models for enzyme engineering; all previous examples have used solved crystal structures as inputs for machine learning approaches. The use of directed evolution for biosensor development, and the use of computational predictions to rapidly exploit the capabilities of the biosensor is overall an important milestone in how to combine technologies to advance pathway engineering, resulting in one of the first improvements in the early steps of AA biosynthesis. To clarify the novelty of our combined approaches, we have made text additions to the Results and Discussion (please refer to Major Point #1)

Reviewer: This reviewer agrees that the new training data adds value and goes beyond-state-of-the-art in computational protein design. The responses in Result and Discussion sections are satisfactory.

“Likewise, no generally applicable mechanistic understanding of the engineered MTase is provided. These

points all limit the scope of the work presented beyond the impactful previous studies from this team.

According to this reviewer, this precludes publishing the study in this journal. This said the optimization of

MutComputeX is very interesting, and so are the novel RamR and enzyme variant, and the study should be suitable for a more specialised journal focusing on method development for metabolic/enzyme engineering or computational biology.”

>> We have since collected structural and kinetic data on the improved Nb4OMT triple mutant. The new X-ray crystal structure, described in the new Figure 6, yield a mechanistic explanation for the key role of the A53M mutation in improving catalytic activity by 4-fold, and an in-depth description of this explanation is found in the new Results section entitled “Crystal structure of an improved norbelladine methyltransferase.” Please again refer to Major Point #1, which fully describes the new Nb4OMT variant’s structure and kinetics.

Reviewer: This reviewer agrees that the new structure and enzyme kinetic data adds mechanistic understanding to the function of the engineered MTase. The updates experimental data and responses in Result and Discussion sections are indeed satisfactory.

Major comments/concerns:

“In the first Result paragraph it states that norbelladine was docked on to RamR, yet both legends in Fig 1c and Suppl. Fig. 1 states 4NB. In the second Result paragraph it (correctly?) states 4NB-based docking to inform library design. Please explain or correct.”

>> We thank this reviewer for bringing this discrepancy to our attention. The Results section was edited to correct this mistake.

Results: “Identifying a biosensor for the branchpoint amaryllidaceae alkaloid 4-O'Methyl-norbelladine To better understand this promiscuous binding activity, 4NB was docked within the ligand binding pocket of RamR using GNINA ... (p. 5)

Reviewer: All good. Happy to help.

“In relation to Fig. 2 it is unclear to this reviewer what difference there is between panel b) and f) when it comes to RFU/OD. In both panels biosensor 4NB2.1 is used yet panel b) has a signal of $0-5 \times 10^4$, while panel f) shows $0-6 \times 10^5$ RFU/OD. Please explain the shift in signal. In relation to these two plots it would also be needed to further elaborate on the operational range of the 4NB2.1 biosensor. In panel b) it seems to saturate at approx 100 μ M 4NB, while in panel f) it is evident that a (significant?) difference can be observed between 100 and 250 μ M.”

>> We greatly appreciate the keen observation made by this reviewer. The difference in relative fluorescence in Figure 2b compared to 2f is due to a difference in the amount of time the cells were

incubated with the ligand; for Figure 2b, cells were incubated with 4NB for 4 hours, the standard assay time we use for biosensor characterization. In Figure 2f, cells were incubated with 4NB for 18 hours, since these are the same conditions employed when the 4NB2 sensor is used to measure enzyme activity

(as in Figure 3; a longer incubation time allows for the enzyme-catalyzed formation of 4NB). This difference also explains the slight change in operational range between Figure 2b and Figure 2f, as pointed out by this reviewer. To clarify this discrepancy, the following text edits were made to the manuscript:

Legend of Figure 2

“... All data was performed in biological triplicate. Cells were cultured for 4 hours with the ligand in b & c, and for 18 hours with the ligand in f & g. Error bars represent the S.D. +/- the mean. ...” (p. 19)

Methods section “Biosensor response assay”

Cultures were grown for an additional 4 h at 37 °C and 250 r.p.m. and subsequently centrifuged (3,500g, 4 °C, 10 min), except in the case of the HPLC comparison assays (Figure 2f,g), where cultures were grown for an additional 18 hours at 37 °C and 250 r.p.m. in order to compare sensor response in fermentation-relevant conditions. (p. 31)

Reviewer: This response and update to the figure legend and Methods section is satisfactory.

“On the note “...it failed to enhance Nb4OMT activity”, from the epPCR variant library screen, could the authors provide speculations or data indicating why the phenotypes from colonies on plates to population/single cell fluorescence from liquid cultures did not correlate? Plate screens are very powerful, and dirt cheap. Readers would benefit from the authors’ considerations here.”

>> We apologize for the confusion. We did not directly compare the fluorescence of colonies on plates to the fluorescence of cell populations in liquid culture. We picked the brightest colonies on plates into liquid culture, but at this stage we were not able to readily quantitatively compare the fluorescence of these colonies to cells expressing the wild-type enzyme. In the subsequent population measurements

with liquid cultures we did compare fluorescence produced by the mutant enzymes to the wild-type enzyme and found no significant difference. In fact, many of our top variants recovered from this screen yielded enzymes with the same exact genotype as the wild-type enzyme. This result highlights the need for computational approaches to more rapidly advance enzyme engineering.

To improve clarity for the reader, we have made the following edit in the results section:

The library of enzyme variants was transformed into cells containing the pSens4NB2.1 plasmid, plated on solid media containing norbelladine, and highly fluorescent colonies were isolated and then individually phenotyped in a secondary, quantitative liquid-based fluorescence screen where they were compared to the wild-type enzyme. Unfortunately, this approach was not able to identify variants that outperformed the wild-type enzyme in the liquid-based screen. (p. 8)

Reviewer: Thank you for the clarification (also, on behalf of your future readers).

“Since epPCR library size of Nb4OMT is not estimated/indicated and that no variants with enhanced activity was identified from the plate assay, it is unclear to this reviewer why the authors move on to adopt machine learning based on structure-guided engineering.”

>> We agree that the rationale for adopting a machine learning-based approach could be clarified to the reader. While the plate-based approach may have eventually yielded improved enzyme variants (as we have shown previously, d'Oelsnitz S, Kim W, et al. 2022), it requires a very idiosyncratic development of libraries and selection pressures, and likely has to be reoptimized for each new enzyme. The power of machine learning is that when it is pre-trained on available enzymes (as MutComputeX is), it ‘pre-learns’ what is likely a good or bad substitution for a given position, thus greatly reducing the number of possibilities that need to be tested, and (with an appropriate biosensor) allowing a rapid screen that may proceed more efficiently than an unoptimized selection. In fact, the machine learning-based approach could find mutations that were unlikely to occur in the error-prone library. For example, the beneficial A53M mutation identified by our AI model required two mutations (GCG → ATG), which would be extremely rare in an epPCR library. Accordingly, we have made the following edits to the results section

to clarify this rationale, and provide detail on the epPCR size:

To improve Nb4OMT activity in a microbial host we initially carried out directed evolution starting from randomly mutagenized libraries, via error-prone PCR, which generated an average of three mutations per gene. (p. 8)

To pursue a complementary approach to enzyme engineering, we sought to use machine learning to guide enzyme design, an approach that could identify variants unlikely to occur via random mutagenesis. (p. 8)

Reviewer: OK. Based on the previous answer from the authors and their answer to this question, this reviewer agrees that it indeed is relevant to “cover-their-loses” from the DE approach and continue with ML-guided predictions. One follow-up question from this: Out of keen interest, if the authors have sequence data from the screen of the epPCR-based library, would it be possible to make a comparison on MutComputeX’s learning performance (e.g. learning curves showing MAE between observations and predictions based on increasing numbers of partitions) with vs without the DE variant data? The outcome could provide data-driven evidence whether the extension of training data based on biosensor-assisted DE adds value to MutComputeX performance, and thereby also better link the whole story. It would also extend nicely from work in F. Arnold’s lab on DE x ML-guided protein engineering. Just a thought for a new Suppl. Figure, not an absolute requirement for publication.

“The plate-screen is claimed to yield different fluorescence intensities (even though this data is not presented), and if this was the case why would colonies from this assay not form relevant training data for ML?”

>> The machine learning models used in this study are self-supervised models, which is a type of unsupervised learning. Thus, the architecture of the model is not structured in a way that could use the colony assay data for the training process.

Reviewer: Ok. Point taken.

“It is an over-statement to say that a structure-based CNN-based approach was pursued when no Nb4OMT structure is available to support the training data. Authors should rather state that CNN was

considered but aborted due to lack of structural data.”

>> We thank this Reviewer for bringing up a potential source of confusion, as the reviewer essentially makes

the case regarding the novelty of our results. It is true that no Nb4OMT structure was solved when pursuing enzyme engineering; however, we were able to create an AlphaFold2 structural model of the Nb4OMT protein docked with SAH and norbelladine, and used this as the input to our model. Therefore, we were able to leverage recent advancements in machine learning structure prediction (AlphaFold2) and ligand docking (GNINA) to “fill in the gap” when no crystal structure data was available; a point we now emphasize in terms of the novelty of our results (Major Point #1). We have also made the following text edits to the Results section.

To produce MutComputeX-guided designs, we generated a Nb4OMT enzyme structure file to serve as an input to the model. Unfortunately, Although the structure of the Nb4OMT enzyme had not been solved, preventing the generation of structure-based CNN predictions for substitutions we were able to create a de novo structural model for Nb4OMT was generated using AlphaFold226, which was then docked with both the S-adenosyl-homocysteine (SAH) cofactor and norbelladine using GNINA1.025. (p. 10).

Reviewer: This makes sense. Thank you for the clarification. The response and update is satisfactory.

Minor comments/concerns:

“L3 in Intro. Please insert what AA stands for - Amaryllidoideae alkaloids?”

>> The following text edit was made in the Introduction to clarify what AA refers to:

Among the approximate ~600 reported Amaryllidoideae alkaloids (AAs), those derived from the lycorine, haemanthamine, and narciclasine scaffolds have been used as lead molecules in anticancer research .

One of the most notable Amaryllidoideae alkaloids (AAs) is galantamine, a selective ... (p. 3)

Reviewer: OK.

“Legend for Suppl. Fig. 2 needs updating on numbering of subpanels. Also, no indication of replicates is provided in the figure legend. Please update this, incl indication of error bars as well as statistical testing if applied.”

>> We greatly appreciate the observation made by this Reviewer. We have since updated the legend for Supplementary Figure 2.

Supplementary Figure 2: The sensitivity and selectivity of RamR mutants evolved for 4-OMe-norbelladine. (a) Dose response measurements and genotypes of generation one RamR sensors. (b) Selectivity of generation one RamR sensors. (c) Dose response measurements and genotypes of generation two RamR sensors. (d) Selectivity of generation two RamR sensors. Experiments were conducted in biological triplicate. Error bars represent the S.D. +/- the mean. (p. 3)

Reviewer: Ok, except still no indication of numbers of replicates (e.g. n = x).

Reviewer #2 (Remarks to the Author):

I would like to thank the authors for addressing the issues I raised during the previous round of review. The authors' revisions have resolved them.

However, the revised version newly includes the kinetic data analysis, and the fit reported in Figure S12 is very poor for all three variants. The three lowest substrate concentrations are completely off in all three variants, most likely leading to a significant overestimation of k_{cat}/K_m . Moreover, in the case of WT but not mutants, these concentrations indicate that the reaction was over after 50 min, which does not seem to be reflected in the fit. My guess is that this is due to the mentioned substrate lost during the reaction due to oxidation and degradation. This must be taken into account in fitting, either as a scaling factor to the observed signal or as a separate competing $S \rightarrow P^*$ reaction, which will prevent KinTek from trying to match the endpoints (at the infinite time) with the initial substrate concentrations. Also, the

three highest substrate concentrations seem to point in the direction of substrate inhibition, which most likely lowered the apparent k_{cat} reported. This inhibition must also be taken into account in the models.

Overall, these observations imply that the reported constants are incorrect, and the kinetic analysis must be redone before the manuscript can be accepted for publication. I recommend checking the Michaelis–Menten plot first to understand if the data quality and model selected are adequate and performing the global fit only after this check.

As a minor issue, the newly added claim in the Discussion, “MutComputeX is the first model that has been explicitly trained to generalize to non-protein atoms, such as nucleic acids and ligands,” is inaccurate and must be clarified. Now, it implies that no model has been trained before to consider non-protein atoms. However, there exists a large domain of protein-ligand binding affinity prediction with a plethora of ML-based models.

Reviewer #3 (Remarks to the Author):

The authors corrected the manuscript and added several results into the revised version of the manuscript. However, the novelty of the manuscript is very limited.

(a) Major point #1. The authors clarified that the largest sources of novelty within this paper come from ... In abstract “engineer amaryllidaceae alkaloid production in *Escherichia coli*.”

4'-O-Methylnorbelladine is the common biosynthetic intermediate of Amaryllidaceae alkaloids (AA). It is not an AA viewing from the chemical structure. The authors corrected the title of the manuscript. Nonetheless, it did not change in the context of the whole manuscript. It can be easy to chemically synthesize 4'-O-methyl-norbelladine with high yield. However, in the manuscript, the authors clarified that a 60% improvement of the yield of 4'-O-methyl-norbelladine, 17-fold reduction of remained substrate, and 3-fold lower of the production of 3'-O-methyl-norbelladine. The readers will find that only ~ 5 μM of 4'-O-methyl-norbelladine was detected in the biotransformation mixture when 500 μM of norbelladine was fed. The authors explained that the substrate norbelladine is subjected to be degraded by oxidation. It cannot be acceptable for the so slow conversion rate. However in Fig 3b, the readers will find ~ 100 μM of 4'-O-methyl-norbelladine when fed with 500 μM of norbelladine. Additionally AI has been applied in many research fields recently. The readers will find that the authors overstated the novelty of the manuscript.

(b) The authors corrected the manuscript and added some new experimental results. However, there are so many errors presented in the revised version of the manuscript. e. g., "4-O'Methylnorbelladine", "norbelladine 4-O'Methyltransferase", "one variant bearing two amino acid substitutions (4NB1.2, K63T and L66M) displayed a 20-fold selectivity for 4NB over norbelladine. Although two other RamR variants had greater sensitivity for 4NB, the higher selectivity of the 4NB2.1 variant rendered it more suitable for accurately monitoring pathway activity", "NMR spectra were taken on the 500 MHZ", "CaCl₂", "Na₂HPO₄", "KH₂PO₄",...

REVIEWERS' COMMENTS

Reviewer #2 (Remarks to the Author):

I want to thank the authors for revisiting their kinetic analysis. I believe it has been improved, and the models are now more consistent with the provided Michaelis-Menten plots. Although the three lowest concentrations still show a small degree of discrepancy, the more cautious wording and reaction rate comparisons in the text are adequate for this type of validation. Therefore, I do not have any more comments, and I recommend accepting the manuscript.